# *Think Fast and Slow:* Step-Level Cognitive Depth Adaptation for LLM Agents

**Ruihan Yang**[1] **Fanghua Ye**[2] **Xiang Wei**[2] **Ruoqing Zhao**[2] **Kang Luo**[2] **Xinbo Xu**[1] **Bo Zhao**[2] **Ruotian Ma**[2]
**Shanyi Wang**[2] **Zhaopeng Tu**[2] **Xiaolong Li**[2] **Deqing Yang**[1] **Liefeng Bo**[2]

## Abstract

Large language models (LLMs) are increasingly deployed as autonomous agents for multi-turn decision-making tasks. However, current agents typically rely on fixed cognitive patterns: non-thinking models generate immediate responses, while thinking models engage in deep reasoning uniformly. This rigidity is inefficient for long-horizon tasks, where cognitive demands vary significantly from step to step, with some requiring strategic planning and others only routine execution. In this paper, we introduce **COGROUTER**, a framework that trains agents to dynamically adapt cognitive depth at each step. Grounded in ACT-R theory, we design four hierarchical cognitive levels ranging from instinctive responses to strategic planning. Our two-stage training approach includes **Cognition-aware Supervised Fine-tuning** (COSFT) to instill stable level-specific patterns, and **Cognition-aware Policy Optimization** (COPO) for step-level credit assignment via confidence-aware advantage reweighting. The key insight is that appropriate cognitive depth should maximize the confidence of the resulting action. Experiments on ALFWorld and ScienceWorld demonstrate that COGROUTER achieves state-of-the-art performance with superior efficiency.

## 1. Introduction

Recent advances in large language models (LLMs) (OpenAI, 2023; Gemini Team, 2023) have enabled their deployment as autonomous agents capable of planning and executing complex multi-turn tasks across domains, including code generation (Yuan et al., 2024; Lu et al., 2025), software en-

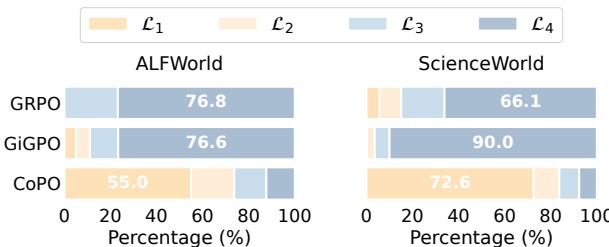

*Figure 1.* Illustration of the **cognitive rigidity issue**: While COPO maintains an adaptive cognitive distribution (bottom), standard RL methods like GRPO (top) collapse to uniform deep thinking ($\mathcal{L}_4$), wasting computational resources on routine steps. $\mathcal{L}_1$–$\mathcal{L}_4$ represent increasing cognitive depth, from instinctive responses to strategic planning.

gineering (Jimenez et al., 2024), web interaction (Yao et al., 2023a), and embodied environments (Yang et al., 2025b). In these settings, agents must continuously perceive observations, reason about actions, and execute decisions over long horizons. Unlike single-turn tasks where a fixed thinking strategy may suffice, agentic environments are highly dynamic and impose **varying cognitive demands** at each step within a task.

Consider the "use thermometer" task in Science-World (Wang et al., 2022), where the agent must first locate the thermometer and then measure the temperature of an unknown substance. In the initial exploration phase, the agent must plan strategically, deciding which rooms to search and in what order to efficiently locate the required objects. When encountering unexpected situations, such as searching the workshop but finding no thermometer, the agent needs to reflect on the failed search and revise its exploration plan. In contrast, many mid-trajectory steps are routine. After executing "open door to bathroom" and observing that the door opens, the natural next action is simply "go to bathroom" without extended thinking. This illustrates that agentic tasks exhibit significant **step-wise heterogeneity** in thinking demands, ranging from instinctive reactions to strategic planning.

However, most agents rely on **fixed cognitive patterns**. Non-thinking models produce reflexive actions at every step, while thinking models (*e.g.*, DeepSeek-R1 (DeepSeek-AI et al., 2025a)) engage in deep chain-of-thought reasoning

[1]Fudan University [2]Tencent Hunyuan Multimodal Department. Correspondence to: Fanghua Ye <fanghuaye@tencent.com>, Zhaopeng Tu <zptu@tencent.com>, Deqing Yang <yangdeqing@fudan.edu.cn>.

*Proceedings of the 43rd International Conference on Machine Learning*, Seoul, South Korea. PMLR 306, 2026. Copyright 2026 by the author(s).

throughout the entire trajectory. This rigidity creates a critical inefficiency: thinking models consume excessive tokens on routine steps where simple responses suffice, while non-thinking models fail on complex decisions requiring strategic planning. A natural solution is to train agents that adaptively allocate cognitive resources. However, as shown in Figure 1, even when initialized with balanced cognitive capabilities, standard reinforcement learning (RL) methods tend to collapse toward uniform deep thinking. While recent work has explored adaptive thinking for single-turn tasks (Lou et al., 2025; Zhang et al., 2025a), these methods typically employ task-level adaptation with binary switching, failing to capture the **step-wise, multi-level** cognitive demands within agentic tasks.

This raises a central challenge: *How can we train agents to adaptively allocate cognitive depth across multiple levels at each step, balancing both task performance and computational efficiency?* To address this, we introduce COGROUTER, a framework grounded in the Adaptive Control of Thought-Rational (ACT-R) theory (Anderson, 1982), which posits that humans dynamically allocate cognitive resources from automatic procedural execution to deliberate reasoning. COGROUTER trains agents to modulate their cognitive depth across **four hierarchical levels**: from instinctive responses ($\mathcal{L}_1$) to strategic planning ($\mathcal{L}_4$).

Training such adaptive allocation is non-trivial due to the sparsity of rewards in agentic environments. We propose a two-stage approach: *Cognition-aware Supervised Fine-tuning* (COSFT) to instill stable, level-specific cognitive patterns, followed by *Cognition-aware Policy Optimization* (COPO). COPO is a novel RL algorithm that enables step-level credit assignment via confidence-aware advantage reweighting. The core insight is that an appropriate cognitive depth should facilitate confident action prediction.

We evaluate COGROUTER on two challenging interactive benchmarks, ALFWorld (Shridhar et al., 2020) and ScienceWorld (Wang et al., 2022). Experimental results demonstrate that COGROUTER achieves state-of-the-art performance with superior token efficiency. Using Qwen2.5-7B as the base model, our approach achieves an 82.3% average success rate, significantly outperforming GPT-4o (+40.3%), OpenAI-o3 (+18.3%), and GRPO (+14.0%), while reducing token consumption by 62% compared to GRPO.

In summary, our contributions are three-fold:

1. We identify and formalize the "cognitive rigidity" issue in LLM agents, illustrating how fixed reasoning patterns lead to inefficiency in long-horizon tasks with heterogeneous step-wise demands.

2. We propose COGROUTER, a framework grounded in ACT-R theory that defines four hierarchical cognitive

levels and utilizes a two-stage training pipeline (COSFT and COPO) to enable dynamic cognitive adaptation.

3. We introduce COPO, a novel RL algorithm that performs step-level credit assignment via confidence-aware advantage reweighting, allowing the model to learn efficient reasoning without collapsing to uniform deep thinking.

## 2. Preliminary

**Partially Observable Markov Decision Process** In this paper, we model the agentic task as a partially observable Markov decision process (POMDP) $\mathcal{M} = (\mathcal{X}, \mathcal{S}, \mathcal{A}, \mathcal{O}, \mathcal{T}, \mathcal{R})$, where $\mathcal{X}$ denotes the set of instructions, $\mathcal{S}$ represents the set of environment states, $\mathcal{A}$ is the action space, and $\mathcal{O}$ is the observation space. The state transition function is $\mathcal{T} : \mathcal{S} \times \mathcal{A} \to \Delta(\mathcal{S})$, and the reward function is $\mathcal{R} : \mathcal{S} \times \mathcal{A} \to \mathbb{R}$. At each time step $t$, the agent with a policy $\pi_\theta$ receives an observation $o_t \in \mathcal{O}$, which is correlated with the current state $s_t \in \mathcal{S}$. The agent maintains an interaction history $\tau_t = \{x, (o_0, a_0), \dots, (o_{t-1}, a_{t-1}), o_t\}$ and samples an action $a_t \sim \pi_\theta(\cdot \mid \tau_t)$, which induces a state transition $s_{t+1} \sim \mathcal{T}(\cdot \mid \tau_t, a_t)$. An episode terminates at step $T$, yielding trajectory $\tau = \{x, (o_0, a_0), \dots, (o_T, a_T)\}$ and terminal reward $\mathcal{R}(\tau)$. The agent's objective is to learn a policy $\pi_\theta$ that maximizes the expected terminal reward:

$$\theta^* = \arg\max_\theta \mathbb{E}_{a_t \sim \pi_\theta(\cdot|\tau_t)}[\mathcal{R}(\tau)].$$

**GRPO for Agentic Tasks** Group Relative Policy Optimization (GRPO) (Shao et al., 2024) is an effective method for training models without the need for a separate critic network. In agentic tasks, GRPO operates at the trajectory level. Given a task query $x \in \mathcal{X}$, the agent samples a set of $G$ complete trajectories $\{\tau^{(i)}\}_{i=1}^G$ using the policy $\pi_\theta$. Each trajectory $\tau^{(i)} = \{x, (o_0^{(i)}, a_0^{(i)}), \dots, (o_T^{(i)}, a_T^{(i)})\}$ begins with the instruction $x$, followed by a sequence of observation-action pairs over $T$ time steps. The trajectory-level reward $R_i = \mathcal{R}(\tau^{(i)})$ measures overall task success. The advantage for each trajectory is computed as

$$A^{(i)} = \frac{R_i - \text{mean}\left(\{R_1, R_2, \dots, R_G\}\right)}{\text{std}\left(\{R_1, R_2, \dots, R_G\}\right)}. \tag{1}$$

This trajectory-level advantage $A^{(i)}$ is uniformly assigned to all steps within the trajectory, serving as the learning signal for policy optimization. The objective function for GRPO is

$$\mathcal{J}_{\text{GRPO}}(\theta) = \mathbb{E}_{\substack{x \sim \mathcal{X}, \\ \{\tau^{(i)}\}_{i=1}^G \sim \pi_{\theta_{\text{old}}}(\cdot|x)}} \left[ \frac{1}{G} \sum_{i=1}^G \frac{1}{|a^{(i)}|} \sum_{t=0}^T \sum_{n=1}^{|a_t^{(i)}|} \right.$$

$$\min\left( r_{t,n}^{(i)}(\theta) A^{(i)}, \text{clip}(r_{t,n}^{(i)}(\theta), 1 - \epsilon, 1 + \epsilon) A^{(i)} \right) \Big]$$

$$- \beta \, \mathbb{D}_{\text{KL}}\left[\pi_\theta \| \pi_{\text{ref}}\right],$$

where $|a^{(i)}| = \sum_{t=0}^{T} |a_t^{(i)}|$ is the total number of action tokens in trajectory $\tau^{(i)}$, and $r_{t,n}^{(i)}(\theta) = \frac{\pi_\theta(a_{t,n}^{(i)}|x, \tau_t^{(i)}, a_{t,<n}^{(i)})}{\pi_{\theta_{\text{old}}}(a_{t,n}^{(i)}|x, \tau_t^{(i)}, a_{t,<n}^{(i)})}$ denotes the importance sampling ratio for the $n$-th token in the action $a_t^{(i)}$.

## 3. Methodology

We propose COGROUTER (Figure 2), a framework for training models to dynamically select the appropriate cognitive level at each step. We first define a set of distinct **cognitive levels**, grounded in ACT-R theory (§ 3.2). Building on this design, we introduce a two-stage training pipeline: *1)* **CoSFT**, which integrates level-specific cognitive patterns into the model's base capabilities (§3.3); and *2)* **CoPO**, which optimizes adaptive cognitive level selection through reinforcement learning with confidence-aware advantage reweighting (§3.4).

### 3.1. Task Formulation

In a standard Partially Observable Markov Decision Process (POMDP), the agent generates actions directly from the interaction history. However, complex agentic tasks often demand deeper reasoning at specific steps. To address this, we introduce $N$ **cognitive levels**, denoted as $\mathcal{L} = \{\mathcal{L}_1, \ldots, \mathcal{L}_N\}$, where each level $\mathcal{L}_i$ represents a distinct depth of reasoning. At each time step $t$, the agent with policy $\pi_\theta$, first selects a cognitive level $l_t \in \mathcal{L}$ based on the interaction history: $l_t \sim \pi_\theta(\cdot \mid \tau_t)$. Given $l_t$, it generates an intermediate thinking process: $th_t \sim \pi_\theta(\cdot \mid \tau_t, l_t)$, and then produces an executable action conditioned on this reasoning: $a_t \sim \pi_\theta(\cdot \mid \tau_t, th_t)$. We denote the complete structured output at step $t$ as $y_t = [l_t, th_t, a_t]$. The objective is to maximize the expected terminal reward:

$$\theta^* = \arg\max_\theta \mathbb{E}_{\substack{l_t \sim \pi_\theta(\cdot|\tau_t),\, th_t \sim \pi_\theta(\cdot|\tau_t, l_t) \\ a_t \sim \pi_\theta(\cdot|\tau_t, th_t)}} [\mathcal{R}(\tau)].$$

This formulation requires the agent to adaptively select an appropriate cognitive level based on the complexity of the current state and to generate corresponding reasoning, balancing depth with efficiency.[1]

### 3.2. Cognitive Level Design

Adaptive Control of Thought-Rational (**ACT-R**) (Anderson, 1982) posits that human cognition operates across a spectrum from automatic procedural execution to effortful declarative reasoning, with processing depth inversely related to cognitive load. Motivated by this framework, we propose **four cognitive levels** for agent decision-making

---

[1]Note that the reward $\mathcal{R}(\tau)$ depends jointly on the cognitive level $l_t$, reasoning process $th_t$, and action $a_t$, and does not necessarily increase with reasoning depth.

that dynamically balance cognitive depth with computational efficiency.

**Level 1 $\mathcal{L}_1$ (Instinctive Response)**: $\mathcal{L}_1$ represents fully autonomous processing, corresponding to ACT-R's procedural stage where compiled production rules execute automatically without working memory engagement (Anderson & Lebiere, 1998). The agent produces *immediate responses* based on learned patterns, with no explicit reasoning structure. This mode optimizes for speed in routine scenarios.

**Level 2 $\mathcal{L}_2$ (Situational Awareness)**: $\mathcal{L}_2$ introduces monitored execution with basic deliberation. The agent assesses the *Current State* and *Available Actions* before responding. This level aligns with ACT-R's goal-directed procedural processing, where situational information is maintained in working memory to guide action selection (Taatgen, 2013).

**Level 3 $\mathcal{L}_3$ (Experience Integration)**: $\mathcal{L}_3$ engages retrospective reasoning by incorporating historical information. Beyond situational assessment, the agent specifies the *Goal*, performs *Reflection* on past actions and outcomes, and integrates experiential insights into decision-making. This corresponds to ACT-R's knowledge compilation stage, where declarative memories are retrieved and consolidated to refine procedural knowledge (Anderson, 1982).

**Level 4 $\mathcal{L}_4$ (Strategic Planning)**: $\mathcal{L}_4$ represents the most cognitively demanding mode, requiring extensive *prospective simulation*. The agent evaluates multiple candidate actions by mentally simulating their future consequences and long-term impact. This level mirrors ACT-R's fully declarative stage, engaging complex problem-solving through chunk retrieval and strategic evaluation (Anderson, 2007).

### 3.3. Cognition-Aware Supervised Fine-tuning

While prior work on controllable reasoning often emphasizes the RL stage, we argue that the SFT stage is equally critical for establishing a solid foundation: *1)* The model should first acquire stable cognitive patterns for each cognitive level to prevent format leakage or mode collapse during subsequent RL training; and *2)* Enforcing a balanced distribution over all cognitive levels during SFT mitigates the model's inherent bias toward certain cognitive patterns, which can otherwise interfere with learning to select levels based on task complexity.

To implement the output $y_t = [l_t, th_t, a_t]$ defined in § 3.1, we introduce a structured format. Each step $t$ includes three components: *1)* the cognitive level, enclosed in `<level>` tags; *2)* the thinking process corresponding to the selected level, enclosed in `<think>` tags; *3)* the final executable action, enclosed in `<action>` tags. Using this structured format, the training data is constructed by augmenting expert trajectories with cognitive annotations as follows:

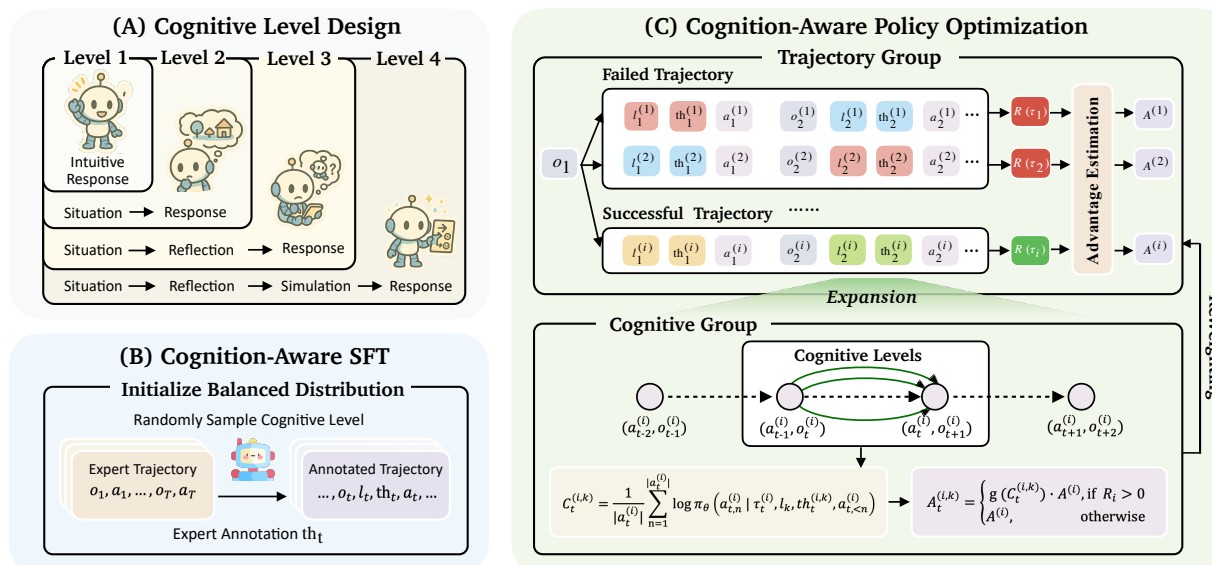

*Figure 2.* Overview of the CoGROUTER framework. We define four cognitive levels $\mathcal{L}_1$–$\mathcal{L}_4$, then introduce a two-stage training process: (1) **CoSFT**, which guides the model to learn stable cognitive patterns across levels with balanced data; (2) **CoPO**, which applies RL with confidence-aware reweighting to help the model adaptively choose suitable levels based on context complexity.

1. We collect a dataset of successful trajectories $\mathcal{T}^* = \{(x, o_0, a_0^*, \ldots, o_T, a_T^*)\}$, comprising only observation-action pairs generated by expert model (*i.e.*, GPT-4o).

2. At each step $t$, we randomly sample a cognitive level $l_t$ to ensure balanced coverage across all levels.

3. Given history $\tau_t^* = \{x, (o_0, a_0^*), \ldots, (o_{t-1}, a_{t-1}^*), o_t\}$ and the ground-truth action $a_t^*$, we prompt the expert model to complete the thinking process $th_t$ at the selected level $l_t$ (see Appendix C for the detailed prompt).

The objective minimizes the negative log-likelihood over the resulting balanced dataset $\mathcal{D}_{\text{cog}} = \{(\tau_t^*, l_t, \text{th}_t, a_t^*)\}_{t=0}^T$, which features an approximately uniform distribution across four cognitive levels:

$$\mathcal{L}_{\text{CoSFT}} = -\mathbb{E}_{\mathcal{D}_{\text{cog}}} \left[ \sum_{t=0}^T \log \pi_\theta \left( y_t \mid \tau_t^* \right) \right],$$

where $\pi_\theta$ is the policy parameterized by $\theta$, and $y_t = [l_t, \text{th}_t, a_t^*]$ denotes the structured output. As an ablation, we also construct an alternative dataset where the expert model directly selects the cognitive level at each step rather than random sampling.

### 3.4. Cognition-Aware Policy Optimization

While group-based RL algorithms (*e.g.*, GRPO) have proven highly effective for training LLMs on reasoning tasks, they assign trajectory-level advantages uniformly across all steps, without distinguishing whether the cognitive pattern at each step is contextually appropriate. To address this, we propose **Cognition-Aware Policy Optimization (CoPO)**, which enables step-level credit assignment based on action prediction confidence (Figure 2). The key insight is that *an appropriate cognitive pattern should facilitate confident and correct action selection.* CoPO modulates step-level advantages using the model's prediction confidence, allowing it to learn which cognitive depth is appropriate for each step. The algorithm is shown in Algorithm 1 in Appendix.

**Reward Design** Given trajectory $\tau^{(i)}$, the trajectory-level reward $R_i$ consists of two components:

$$R_i = R_i^{\text{task}} \times R_i^{\text{format}},$$

where $R_i^{\text{task}} \in \{0, 1\}$ indicates task success, and $R_i^{\text{format}} \in \{0, 1\}$ indicates whether all actions in the trajectory conform to the required structured format (*i.e.*, properly using `<level>`, `<think>`, and `<action>` tags). We set $R_i^{\text{format}} = 0$ if any action deviates from this format, imposing a strict penalty to enforce format consistency.

**Cognitive Group Construction** For each successful trajectory ($R_i > 0$), we expand the training data by constructing a **cognitive group** at each step. Specifically, using the current policy $\pi_\theta$, we regenerate the thinking process under all four cognitive levels while keeping the observation $o_t^{(i)}$ and action $a_t^{(i)}$ fixed. This yields the cognitive group $e_t^{(i)} = \{e_t^{(i,k)}\}_{k=1}^4$, where each variant is defined as $e_t^{(i,k)} = [l_k, th_t^{(i,k)}, a_t^{(i)}]$, corresponding to cognitive level $l_k$. Importantly, all variants produce the same action $a_t^{(i)}$,

but differ in the thinking process that leads to it. To assess the suitability of each cognitive level, we evaluate how confidently the model predicts the action when conditioned on each thinking process:

$$C_t^{(i,k)} = \frac{1}{|a_t^{(i)}|} \sum_{n=1}^{|a_t^{(i)}|} \log \pi_\theta \left( a_{t,n}^{(i)} \mid \tau_t^{(i)}, l_k, th_t^{(i,k)}, a_{t,<n}^{(i)} \right),$$

where $a_{t,n}^{(i)}$ is the $n$-th token in the action $a_t^{(i)}$. Higher confidence scores indicate stronger alignment between the thinking process and the resulting action.

**Confidence-Aware Advantage Reweighting** To compare cognitive levels within each group, we normalize the confidence scores:

$$C_{\text{norm},t}^{(i,k)} = \frac{C_t^{(i,k)} - \mu_t^{(i)}}{\sigma_t^{(i)}}, \tag{2}$$

where $\mu_t^{(i)}$ and $\sigma_t^{(i)}$ are the mean and standard deviation of confidence scores within the cognitive group $e_t^{(i)}$. We then apply a temperature-scaled softmax to convert these normalized scores into relative weights:

$$g(C_t^{(i,k)}) = \frac{\exp(m \cdot C_{\text{norm},t}^{(i,k)})}{\sum_{j=1}^{4} \exp(m \cdot C_{\text{norm},t}^{(i,j)})}, \tag{3}$$

where $m$ is a temperature that controls the sharpness of the distribution[2]. Since $\sum_{k=1}^{4} g(C_t^{(i,k)}) = 1$ by construction, these weights redistribute the total advantage across cognitive levels without changing its magnitude. These weights are used to modulate the step-level advantage:

$$A_t^{(i,k)} = \begin{cases} g(C_t^{(i,k)}) \cdot A^{(i)}, & \text{if } R_i > 0 \\ A^{(i)}, & \text{otherwise} \end{cases} \tag{4}$$

where $A^{(i)}$ is the trajectory-level advantage (Eq. 1). For successful trajectories ($R_i > 0$), this reweighting amplifies advantages for cognitive levels that facilitate confident action predictions, while attenuating uncertain ones. For failed trajectories, no cognitive group is constructed, and the advantage remains at the trajectory level.

**CoPO Optimization** For successful trajectories ($i \in \mathcal{I}^+$), we use the cognitive groups $e_t^{(i)}$ with reweighted advantages $A_t^{(i,k)}$ for each level from Equation 4. For failed trajectories ($i \in \mathcal{I}^-$), we retain only the original structured output $y_t^{(i)}$ with trajectory-level advantage $A^{(i)}$. The objec-

---

[2]We set $m = 2$ in our experiments.

tive function is

$$\mathcal{J}_{\text{CoPO}}(\theta) = \mathbb{E}_{\substack{x \sim \mathcal{X} \\ \{\tau^{(i)}\}_{i=1}^G \sim \pi_{\theta_{\text{old}}}(\cdot|x)}} \left[ \frac{1}{G} \left( \sum_{i \in \mathcal{I}^+} \frac{1}{|e^{(i)}|} \sum_{t=0}^{T} \sum_{k=1}^{4} \right. \right.$$
$$\sum_{n=1}^{|e_t^{(i,k)}|} \min \left( r_{t,n}^{(i,k)} A_t^{(i,k)}, \overline{r}_{t,n}^{(i,k)} A_t^{(i,k)} \right)$$
$$\left. \left. + \sum_{i \in \mathcal{I}^-} \frac{1}{|y^{(i)}|} \sum_{t=0}^{T} \sum_{n=1}^{|y_t^{(i)}|} \min \left( r_{t,n}^{(i)} A^{(i)}, \overline{r}_{t,n}^{(i)} A^{(i)} \right) \right) \right]$$
$$- \beta \, \mathbb{D}_{\text{KL}}[\pi_\theta \| \pi_{\text{ref}}], \tag{5}$$

where $|e^{(i)}| = \sum_{t,k} |e_t^{(i,k)}|$ and $|y^{(i)}| = \sum_t |y_t^{(i)}|$ are the total tokens for normalization, $r_{t,n}^{(i,k)} = \frac{\pi_\theta(e_{t,n}^{(i,k)}|x,\tau_t^{(i)},e_{t,<n}^{(i,k)})}{\pi_{\theta_{\text{old}}}(e_{t,n}^{(i,k)}|x,\tau_t^{(i)},e_{t,<n}^{(i,k)})}$ and $r_{t,n}^{(i)} = \frac{\pi_\theta(y_{t,n}^{(i)}|x,\tau_t^{(i)},y_{t,<n}^{(i)})}{\pi_{\theta_{\text{old}}}(y_{t,n}^{(i)}|x,\tau_t^{(i)},y_{t,<n}^{(i)})}$ denote the importance sampling ratios for successful and failed trajectories respectively, and $\overline{r} = \text{clip}(r, 1-\epsilon, 1+\epsilon)$ is the clipped ratio.

## 4. Experiments

### 4.1. Experimental Setup

We conduct experiments on two challenging environments **ALFWorld** (Shridhar et al., 2020) and **Science-World** (Wang et al., 2022). Further details can be found in Appendix B. Following prior work (Zhang et al., 2025b), we evaluate models on the test sets of both environments: 200 simulations for ALFWorld and the same for ScienceWorld. For ALFWorld, we report the success rate (**SR**) for each task category and overall. For ScienceWorld, we use both the average final score and success rate as the evaluation metrics. We also compute the average output token usage per task (**#Tokens**) as a measure of model efficiency.

We compare with several baselines: *1)* **Prompt-based methods**, including ReAct (Yao et al., 2023b) and Reflexion (Shinn et al., 2023), a self-refinement method that summarizes each trial to guide future decisions. We also evaluate R1-distilled variants of Llama3.1-8B and Qwen2.5-7B, prompted with ReAct; *2)* **Training-based methods**, covering both offline and online methods. Offline methods include SFT and ETO (Song et al., 2024). We also include CoSFT (§3.3), which uniformly samples cognitive levels, and its variant CoSFT_{exp}, which uses expert-selected levels. Online methods include GRPO (Shao et al., 2024), GiGPO (Feng et al., 2025), and AdaptThink (Zhang et al., 2025a). We also compare with several non-thinking and thinking frontier models (see Appendix H for details).

**Training Settings** We use Llama3.1-8B (Meta, 2024) and Qwen2.5-7B (Yang et al., 2024a) as base models, with training data drawn from ScienceWorld (2,120 simulations) and

*Table 1.* Performance comparison across different methods on ALFWorld and ScienceWorld.

| Method | ALFWorld | | ScienceWorld | | | Average | |
|---|---|---|---|---|---|---|---|
| | SR | #Tokens | Score | SR | #Tokens | SR | #Tokens |
| **Frontier Large Models** | | | | | | | |
| gpt-4o-2024-08-06 (OpenAI, 2025a) | 61.5 | 860.8 | 57.0 | 22.5 | 1009.9 | 42.0 | 935.4 |
| deepseek-v3-0324 (DeepSeek-AI et al., 2025b) | 52.0 | 1171.2 | 13.0 | 6.0 | 1013.4 | 29.0 | 1092.3 |
| openai-o3 (OpenAI, 2025b) | 74.0 | 4290.9 | 72.4 | 54.0 | 5184.0 | 64.0 | 4737.5 |
| deepseek-r1-0528 (DeepSeek-AI et al., 2025a) | 73.0 | 6230.9 | 58.9 | 40.0 | 16502.7 | 56.5 | 11366.8 |
| gemini-2.5-pro-0506 (Gemini Team, 2025) | 70.0 | 4896.5 | 65.2 | 44.0 | 8942.4 | 57.0 | 6919.5 |
| **Qwen2.5-7B-Instruct** | | | | | | | |
| ReAct (Yao et al., 2023b) | 50.5 | 666.4 | 8.0 | 2.5 | 1645.1 | 26.5 | 1155.8 |
| + R1-Distill | 4.5 | 33620.5 | 2.5 | 0.5 | 21221.9 | 2.5 | 27421.2 |
| Reflexion (Shinn et al., 2023) | 42.5 | 2340.8 | 10.1 | 1.0 | 3396.8 | 21.8 | 2868.8 |
| SFT | 62.5 | 494.1 | 54.1 | 19.0 | 660.8 | 40.8 | 577.5 |
| ETO (Song et al., 2024) | 69.5 | 540.6 | 53.9 | 21.5 | 694.3 | 45.5 | 617.4 |
| AdaptThink (Zhang et al., 2025a) | 81.5 | 1372.1 | 62.5 | 44.5 | 1314.2 | 63.0 | 1343.2 |
| CoSFT$_{exp}$ | 78.5 | 3151.0 | 61.2 | 29.5 | 3973.8 | 54.0 | 3562.4 |
| CoSFT | 57.0 | 3912.7 | 61.1 | 35.5 | 3643.4 | 46.3 | 3778.1 |
| + GRPO (Shao et al., 2024) | 83.5 | 4994.6 | 71.1 | 53.0 | 3739.9 | 68.3 | 4367.3 |
| + GiGPO (Feng et al., 2025) | 88.0 | 2955.5 | 67.3 | 47.0 | 4602.8 | 67.5 | 3779.2 |
| + CoPO (Ours) | **92.5** | 1739.4 | **84.6** | **72.0** | 1543.4 | **82.3** | 1641.4 |
| **Llama3.1-8B-Instruct** | | | | | | | |
| ReAct (Yao et al., 2023b) | 13.0 | 959.4 | 13.7 | 3.0 | 1268.1 | 8.0 | 1113.8 |
| + R1-Distill | 2.5 | 21720.8 | 2.2 | 0.0 | 6940.8 | 1.3 | 14330.8 |
| Reflexion (Shinn et al., 2023) | 18.5 | 3645.8 | 17.8 | 2.5 | 5072.4 | 10.5 | 4359.1 |
| SFT | 54.5 | 632.0 | 59.9 | 24.0 | 906.6 | 39.3 | 769.3 |
| ETO (Song et al., 2024) | 65.0 | 730.0 | 62.5 | 38.5 | 784.2 | 51.8 | 757.1 |
| AdaptThink (Zhang et al., 2025a) | 79.5 | 1607.2 | 52.3 | 45.5 | 1666.8 | 62.5 | 1637.0 |
| CoSFT$_{exp}$ | 82.0 | 2903.2 | 73.3 | 49.5 | 4588.9 | 65.8 | 3746.1 |
| CoSFT | 55.5 | 3546.6 | 56.4 | 26.0 | 3728.0 | 40.8 | 3637.3 |
| + GRPO (Shao et al., 2024) | 84.0 | 3941.1 | 75.1 | 50.5 | 5757.6 | 67.3 | 4849.4 |
| + GiGPO (Feng et al., 2025) | 89.5 | 6112.0 | 78.2 | 64.0 | 4651.3 | 76.8 | 5381.7 |
| + CoPO (Ours) | **91.5** | 811.6 | **83.7** | **70.5** | 963.8 | **81.0** | 887.7 |

ALFWorld (2,420 simulations). For COSFT, we randomly sample 500 scenarios, collect expert trajectories using GPT-4o (OpenAI, 2025a), and prompt it to complete thinking processes at randomly sampled cognitive levels (§ 3.3). For COPO, we use the remaining simulations, sample different 16 groups per rollout with a group size of 8, and run 150 iterations. Additional details are provided in Appendix E.

### 4.2. Main Results

**COGROUTER achieves state-of-the-art performance with superior token efficiency.** As shown in Table 1, our method, COPO, sets a new state of the art on both ALF-World and ScienceWorld across two different base models. With Qwen2.5-7B, COPO achieves a 92.5% success rate on ALFWorld and 72.0% on ScienceWorld (average 82.3%), substantially outperforming powerful thinking models like openai-o3 (+18.3% average). Crucially, this performance is achieved with remarkable efficiency. Compared to the strong GRPO baseline, COPO uses 62% fewer tokens

on average while delivering a 14.0% higher success rate, directly validating our central claim of balancing effectiveness and computational cost. The results with Llama3.1-8B show a similar trend, confirming the general applicability and robustness of our approach.

**COPO significantly outperforms other RL methods by enabling step-level credit assignment.** When initializing from the same COSFT checkpoint, the superiority of COPO becomes evident. Trajectory-level methods like GRPO and GiGPO improve performance over the SFT baseline at the cost of increasing token consumption significantly. For instance, with Llama3.1-8B, GRPO increases token usage to 3,941.1 on ALFWorld. In contrast, COPO not only achieves the highest success rates (+7.5% over GRPO on ALFWorld) but also dramatically reduces token usage by 79% (to 811.6 tokens). This is because its confidence-aware advantage reweighting provides a fine-grained, step-level signal that teaches the agent to select the leanest cognitive level suffi-

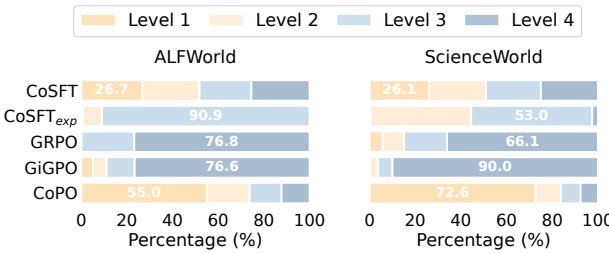

*Figure 3.* Cognitive level distribution. All RL methods (GRPO, GiGPO, CoPO) are initialized from CoSFT. While GRPO and GiGPO collapse to $\mathcal{L}_4$ thinking, CoPO *learns adaptive allocation.*

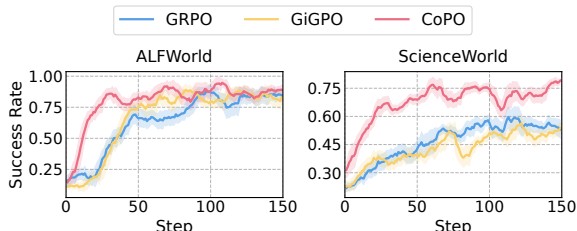

*Figure 4.* Training curves of success rate across RL iterations for GRPO, GiGPO and CoPO on ALFWorld and ScienceWorld. CoPO *achieves faster convergence to higher success rates.*

cient for a confident action, rather than uniformly rewarding all steps in a successful trajectory.

**CoPO prevents cognitive level collapse through confidence-aware credit assignment.** RL baselines struggle to adapt cognitive depth at the step level (Figure 3). Despite starting with a balanced CoSFT initialization, GRPO and GiGPO rapidly collapse to predominantly $\mathcal{L}_4$ thinking (*e.g.*, 76.8% and 76.6% on ALFWorld with Qwen2.5-7B). This stems from coarse credit assignment that indiscriminately rewards deep thinking because it often correlates with higher final rewards, instead of assessing contextual utility. CoPO avoids this collapse via confidence-aware advantage reweighting (§3.4), which evaluates each cognitive level based on the model's action prediction confidence. This enables step-wise adaptive thinking and maintains a balanced cognitive profile (*e.g.*, 55.0% $\mathcal{L}_1$, 18.7% $\mathcal{L}_2$, 14.0% $\mathcal{L}_3$, 12.3% $\mathcal{L}_4$ on ALFWorld). Consequently, CoPO converges much faster (Figure 4): with Qwen2.5-7B, it reaches 90% success on ALFWorld within 100 steps (GRPO: 83.5% at 150 steps) and 76.6% on ScienceWorld by step 80 (GRPO: 59.7%). This demonstrates that confidence-aware reweighting provides more precise, step-level learning signals than trajectory-level baselines. Extended results with Llama3.1-8B are provided in Appendix F.1.

### 4.3. Quantitative Analysis

While Section 4.2 shows CoPO's superior performance and efficiency, its underlying mechanism remains unclear. To

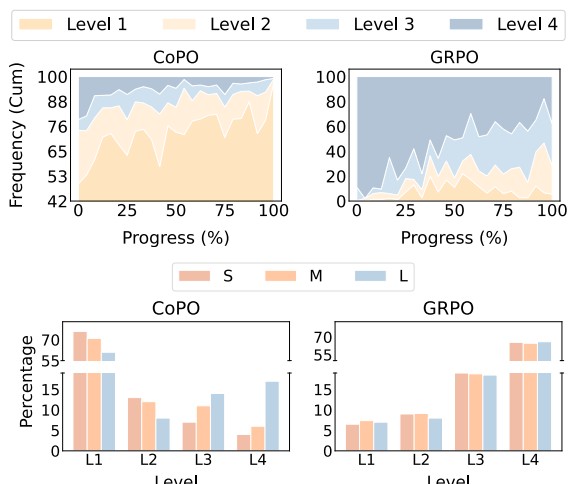

*Figure 5.* Cognitive level distributions across trajectory progress (upper) and task complexity (bottom) on ScienceWorld.

reveal the underlying mechanism under CoPO's superior performance, we analyze the cognitive level distributions along two dimensions: *1) Trajectory-level*: how cognitive patterns evolve across trajectory stages, and *2) Task-level*: how cognitive depth scales with task complexity.

**CoPO learns to allocate cognitive resources dynamically based on the trajectory stage.** We examine how the four cognitive levels are distributed across task progress in ScienceWorld, using Qwen2.5-7B (Figure 5, upper panel). CoPO exhibits distinct, stage-aware patterns. Strategic planning ($\mathcal{L}_4$) peaks at initialization (21.9%) when the agent must perform global goal evaluation and long-horizon planning, then declines to 6.8% as the task structure becomes clearer. Situational awareness ($\mathcal{L}_2$) also features prominently in early stages (26.3%), where parsing observations is critical, before stabilizing around 10%. Conversely, instinctive responses ($\mathcal{L}_1$) surge from 48.5% to over 80% in later stages, reflecting a shift to routine execution. Reflection ($\mathcal{L}_3$) maintains a relatively low and uniform presence, emerging contextually when error correction is needed. This structured allocation is cognitively plausible and highly efficient, directly validating the paper's central hypothesis of step-wise heterogeneity in cognitive demands. A detailed case study is provided in Appendix G.

**CoPO adapts cognitive depth proportionally to task complexity.** To investigate how cognitive levels adapt across tasks of varying difficulty, we compare CoPO and GRPO on ScienceWorld (Figure 5, bottom panel). We categorize tasks into Short (S), Medium (M), and Long (L) based on oracle trajectory lengths (see Appendix B). CoPO demonstrates clear complexity-aware adaptation. As task difficulty increases from S to L, the use of $\mathcal{L}_4$ rises from 7.6% to 22.4% and $\mathcal{L}_3$ from 8.6% to 15.2%, reflecting a

*Table 2.* Ablation studies of key components in CoPO on confidence metrics and training strategies. SR: Success Rate (%). $\mathcal{L}_1$-$\mathcal{L}_4$: Cognitive level distribution (%).

| Variant | SR | #Tokens | $\mathcal{L}_1$ | $\mathcal{L}_2$ | $\mathcal{L}_3$ | $\mathcal{L}_4$ |
|---|---|---|---|---|---|---|
| **Confidence Metric Ablation** | | | | | | |
| CoPO (Ours) | **72.0** | **1543.4** | 72.6 | 11.2 | 8.6 | 7.6 |
| w/ *Max Probs* | 69.5 | 2167.5 | 68.3 | 15.2 | 9.5 | 7.1 |
| w/ *Min Probs* | 67.5 | 2063.6 | 69.5 | 12.7 | 10.7 | 7.1 |
| w/ *Entropy* | 70.0 | 1873.9 | 72.0 | 12.7 | 8.6 | 6.7 |
| **Training Strategy Ablation** | | | | | | |
| w/o *Cold Start* | 64.5 | 3750.6 | 0.0 | 27.1 | 35.5 | 37.3 |
| w/ COSFT$_{\text{expert}}$ | 62.0 | 4383.0 | 0.0 | 24.3 | 69.1 | 0.6 |
| w/ *Corr & Incorr* | 66.0 | 1755.6 | 77.6 | 9.9 | 6.9 | 5.6 |

greater need for strategic planning and reflection. Concurrently, reliance on $\mathcal{L}_1$ decreases from 72.6% to 55.4%. This graduated allocation demonstrates that CoPO successfully learns to scale its cognitive effort with task demands.

### 4.4. Ablation Study

To validate the key design choices of COGROUTER, we conduct ablation studies on: *1)* the confidence metric for advantage reweighting in CoPO, and *2)* training strategies for cognitive level distribution. Results with Qwen2.5-7B on ScienceWorld are presented in Table 2. We also conduct the same ablations on Llama3.1-8B (see Appendix F.3).

**Average log-probability is the most effective confidence metric for assessing cognitive suitability.** We evaluate our choice of average log-probability (§3.4) as the confidence metric by comparing it with three alternatives. Our chosen metric yields the best performance in both success rate and token efficiency. Alternative metrics distort the credit signal. *Min Probs* performs poorly (67.5% SR) because a single low-probability token creates a noisy, overly pessimistic signal. *Entropy* also degrades performance (70.0% SR), as it measures general vocabulary uncertainty rather than targeted confidence in the chosen action. *Max Probs* achieves moderate results (69.5% SR) but overweights local peaks while ignoring the action's overall coherence. These results validate that average log-probability provides the most robust and holistic signal for guiding step-level cognitive selection in CoPO.

**Balanced initialization and success-only reweighting are crucial for effective adaptation.** We examine two critical design decisions in our training pipeline. A balanced SFT initialization proves essential: the w/o *Cold Start* variant fails to establish stable cognitive patterns, leading to distribution collapse (0.0% $\mathcal{L}_1$) and degraded performance (64.5%

SR). Similarly, initializing from a biased expert model (w/ COSFT$_{\text{expert}}$) causes the agent to inherit the teacher's preference, collapsing predominantly to $\mathcal{L}_3$ (69.1%) and preventing it from learning to adapt effectively. Both variants confirm that a balanced SFT stage is vital for enabling effective exploration during RL. Additionally, applying reweighting to both correct and incorrect trajectories (w/ *Corr & Incorr*) creates a perverse incentive. The model learns to evade large penalties for confident mistakes by defaulting to the least confident level, $\mathcal{L}_1$ (77.6%), which sacrifices task performance (66.0% SR) to minimize penalties rather than maximize task success. This confirms that our strategy of rewarding confidence exclusively on successful paths is key to learning effective and adaptive cognitive allocation.

## 5. Related Work

**Interactive and Agentic Environments.** Recent benchmarks evaluate language agents in complex, goal-driven settings, ranging from code generation (Guo et al., 2024) and embodied intelligence (Xi et al., 2024) to social deduction (Yang et al., 2024b; 2025c) and emotional intelligence (Chen et al., 2024). Unlike single-turn tasks, these environments are characterized by long horizons, sparse terminal rewards, and heterogeneous step-wise complexity, motivating the need for adaptive cognitive allocation. We propose step-level cognitive allocation grounded in ACT-R theory (Anderson, 1982), introducing four hierarchical cognitive levels and confidence-aware credit assignment for fine-grained adaptation.

**Hybrid and Adaptive Reasoning.** Large reasoning models such as OpenAI-o1 (OpenAI, 2025b), DeepSeek-R1 (DeepSeek-AI et al., 2025a), and Gemini-2.0-Flash-Thinking (Choi & Na, 2024) employ extended chain-of-thought to solve complex problems. To improve efficiency, recent hybrid thinking approaches (Zhang et al., 2025a; Yang et al., 2025a) dynamically toggle between reasoning and non-reasoning modes. However, these methods typically operate at the trajectory level with binary switching, failing to capture the fine-grained, step-wise cognitive demands of agentic tasks. We propose a hierarchical step-level allocation grounded in ACT-R theory (Anderson, 1982).

**RL for Language Models.** RL has been widely adopted for LLM alignment (Dang et al., 2024) and reasoning enhancement (Hu et al., 2025). Group-based methods like GRPO (Shao et al., 2024), Dr.GRPO (Liu et al., 2025), and DAPO (Yu et al., 2025) estimate advantages without critics but apply uniform credit assignment, which is suboptimal for multi-turn settings. While GiGPO (Feng et al., 2025) groups steps by state similarity, it lacks cognitive awareness. We propose CoPO, which leverages action prediction confidence for precise, step-level credit assignment.

# 6. Conclusion

In this study, we addressed the inefficiency of fixed cognitive patterns in long-horizon agentic tasks. We introduced COGROUTER, a framework grounded in ACT-R theory that enables agents to dynamically allocate cognitive depth across four levels, from instinctive responses to strategic planning. Through COSFT and COPO, we demonstrated that agents can learn to match their reasoning depth to step complexity. Our COPO algorithm effectively solves step-level credit assignment by leveraging action prediction confidence, preventing the mode collapse observed in standard trajectory-level RL methods. Experiments on ALFWorld and ScienceWorld confirm that COGROUTER achieves state-of-the-art performance while drastically reducing token consumption compared to uniform reasoning models. This work highlights the critical role of adaptive cognitive allocation in building efficient and effective language agents.

## Acknowledgement

We appreciate the support from the Chinese NSF General Program (No.62572129). We also acknowledge the use of an icon from Flaticon[3] and thank its creators for providing this visually appealing design.

## Impact Statement

This paper presents a framework for training LLM agents to adaptively allocate cognitive depth, improving both task performance and computational efficiency in interactive environments. There are many potential societal consequences of our work, none which we feel must be specifically highlighted here.

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

# A. Notations

Table 3 summarizes the core notations used in this paper.

*Table 3.* Summary of Notations

| Symbol | Task Formulation |
| --- | --- |
| | **Meaning** |
| $\mathcal{M}$ | Partially Observable Markov Decision Process (POMDP) tuple |
| $\tau$ | Interaction trajectory containing instructions, observations, and actions |
| $\pi_\theta$ | The language agent policy parameterized by $\theta$ |
| $\mathcal{R}$ | Reward function evaluating task success |

| Symbol | Cognitive Modeling |
| --- | --- |
| | **Meaning** |
| $\mathcal{L}$ | Set of four cognitive levels $\{\mathcal{L}_1, \mathcal{L}_2, \mathcal{L}_3, \mathcal{L}_4\}$ derived from ACT-R theory |
| $l_t$ | Selected cognitive level at time step $t$, where $l_t \in \mathcal{L}$ |
| $th_t$ | Internal thinking process generated corresponding to $l_t$ |
| $y_t$ | Structured output at step $t$, defined as $y_t = [l_t, th_t, a_t]$ |
| $\mathcal{D}_{\text{cog}}$ | Cognition-aware dataset used for the SFT stage |

| Symbol | CoPO Optimization |
| --- | --- |
| | **Meaning** |
| $A^{(i)}$ | Trajectory-level advantage calculated via standard GRPO |
| $e_t^{(i)}$ | Cognitive group consisting of counterfactual reasoning paths at step $t$ |
| $C_t^{(i,k)}$ | Action prediction confidence under cognitive level $k$ |
| $g(\cdot)$ | Temperature-scaled softmax function for weight normalization |
| $A_t^{(i,k)}$ | Confidence-aware step-level advantage for fine-grained credit assignment |

# B. Task Details

We evaluate our framework on two challenging agentic environments: **ALFWorld** and **ScienceWorld**. These benchmarks assess the agent's ability to perform long-horizon planning, reasoning, and environment interaction.

**ALFWorld**    ALFWorld (Shridhar et al., 2020) is a household environment built on TextWorld, where agents must explore rooms and apply commonsense reasoning to complete tasks. The action space includes operations such as picking up and placing items, observing the environment, and interacting with furniture. The environment provides feedback based on a set of predefined logical rules. ALFWorld defines six types of tasks: Pick & Place, Examine in Light, Clean & Place, Heat & Place, Cool & Place, and Pick Two & Place. We use the success rate as the evaluation metric. The maximum number of interaction steps is set to 30 for evaluation.

**ScienceWorld**    ScienceWorld (Wang et al., 2022) is a benchmark environment designed to evaluate agents' scientific reasoning abilities, based on a standard elementary science curriculum. It comprises 30 task types, including using measurement instruments and conducting mechanics experiments. The action space is task-specific, and the simulator returns the effects of each action. Following prior work (Lin et al., 2023), we categorize tasks into three groups based on the average length of oracle trajectories: *Short* ($^*Len \leq 20$, avg. 11.76 steps), *Medium* ($20 < {}^*Len \leq 50$, avg. 28.58 steps), and *Long* ($^*Len > 50$, avg. 94.30 steps). These lengths reflect the oracle's optimal paths. We use task score as the primary evaluation metric and report the success rate as a complementary measure. The maximum number of interaction steps is limited to 100 for evaluation.

# C. Instruction Prompt Examples

The system prompts for two agentic environments are presented in Listing 1. The output format for our CogRouter framework introduced in §3.2 is presented in Listing 2.

*Listing 1.* Prompt details for ScienceWorld and ALFWorld.

```
ALFWorld Instruction:
Interact with a household to solve a task. Imagine you are an intelligent agent in a
household environment and your target is to perform actions to complete the task goal. At
the beginning of your interactions, you will be given the detailed description of the
current environment and your goal to accomplish. For each of your turn, you will be given
a list of actions which you can choose one to perform in this turn. After each turn, the
environment will give you immediate feedback based on which you plan your next few steps.
If the environment outputs 'Nothing happened', that means the previous action is invalid
and you should try more options.
Reminder: the action must be chosen from the given available actions. Any actions except
provided available actions will be regarded as illegal.

Your current task is: {task_description}

Prior to this step, you have already taken {step_count} step(s). Below are the most recent
 {history_length} observations and the corresponding actions you took:
{action_history}
You are now at step {current_step} and your current observation is:
{current_observation}
Your admissible actions of the current situation are: {admissible actions}

Now it's your turn to generate next step response.

ScienceWorld Instruction:
You are an agent for science world. Every round I will give you an observation, you have
to respond an action based on the observation to finish the given task.

Here are the actions you may take:
[
{{"action": "open OBJ", "description": "open a container"}},
{{"action": "close OBJ", "description": "close a container"}},
{{"action": "activate OBJ", "description": "activate a device"}},
{{"action": "deactivate OBJ", "description": "deactivate a device"}},
{{"action": "connect OBJ to OBJ", "description": "connect electrical components"}},
{{"action": "disconnect OBJ", "description": "disconnect electrical components"}},
{{"action": "use OBJ [on OBJ]", "description": "use a device/item"}},
{{"action": "look around", "description": "describe the current room"}},
{{"action": "look at OBJ", "description": "describe an object in detail"}},
{{"action": "look in OBJ", "description": "describe a container's contents"}},
{{"action": "read OBJ", "description": "read a note or book"}},
{{"action": "move OBJ to OBJ", "description": "move an object to a container"}},
{{"action": "pick up OBJ", "description": "move an object to the inventory"}},
{{"action": "put down OBJ", "description": "drop an inventory item"}},
{{"action": "pour OBJ into OBJ", "description": "pour a liquid into a container"}},
{{"action": "dunk OBJ into OBJ", "description": "dunk a container into a liquid"}},
{{"action": "mix OBJ", "description": "chemically mix a container"}},
{{"action": "go to LOC", "description": "move to a new location"}},
{{"action": "eat OBJ", "description": "eat a food"}},
{{"action": "flush OBJ", "description": "flush a toilet"}},
{{"action": "focus on OBJ", "description": "signal intent on a task object"}},
{{"action": "wait", "description": "take no action for 10 iterations"}},
{{"action": "wait1", "description": "take no action for 1 iteration"}},
{{"action": "task", "description": "describe current task"}},
{{"action": "inventory", "description": "list your inventory"}}
]

Your current task is: {task_description}
```

```
Prior to this step, you have already taken {step_count} step(s). Below are the most recent
 {history_length} observations and the corresponding actions you took:
{action_history}
You are now at step {current_step} and your current observation is:
{current_observation}

Now it's your turn to generate next step response.
```

*Listing 2.* The output format for CogRouter.

```
There are four thinking levels:
Level 1 - Instinctive Response: Immediate reaction based on intuition, no analysis.
Level 2 - Situational Awareness: Assess current state and available actions before acting.
Level 3 - Experience Integration: Reflect on past actions and outcomes to inform current
decisions.
Level 4 - Strategic Planning: Assess the task goal, past lessons, and current state to
analyze the future impact of each candidate action and optimize the decision.
You must first choose an appropriate level of thinking (one of the four levels) to respond
 based on the given scenario. The chosen level MUST be enclosed within <level></level>
tags.
Next, reason step-by-step using the chosen thinking level. This reasoning process MUST be
enclosed within <think></think> tags.
For Level 1 (Instinctive Response), use the fixed text: "Okay, I think I have finished
thinking." For Levels 2-4, provide detailed reasoning as shown in examples. Once you've
finished your reasoning, you should choose an admissible action for current step and
present it within <action></action> tags.

[Output Format]
Your output must adhere to the following format:
EXAMPLE 1:
<level>1</level>
<think>Okay, I think I have finished thinking.</think>
<action>your_next_action</action>

EXAMPLE 2:
<level>2</level>
<think>
Current state: [Current state and inventory]
Available actions: [What actions are valid right now]
Reasoning: [Choose the best action and explain why]
</think>
<action>your_next_action</action>

EXAMPLE 3:
<level>3</level>
<think>
Goal: [What needs to be accomplished]
Current state: [Current state and inventory]
Available actions: [What actions are valid right now]
Reflection: [How effective were recent actions, what was learned]
Reasoning: [Choose the best action based on experience]
</think>
<action>your_next_action</action>

EXAMPLE 4:
<level>4</level>
<think>
Goal: [What needs to be accomplished]
Current state: [Current state and inventory]
Available actions: [What actions are valid right now]
Reflection: [How effective were recent actions, what was learned]
Evaluation: [Assess the potential effectiveness of each candidate action]
Reasoning: [Choose the optimal action with strategic reasoning]
</think>
<action>your_next_action</action>
```

# D. Algorithm

---

**Algorithm 1** Cognition-Aware Policy Optimization (CoPO)

---

1: **Initialize:** Policy $\pi_\theta$.
2: **for** each training iteration **do**
3:     Collect a batch of trajectories $\mathcal{B} = \{\tau^{(i)}\}$ by running policy $\pi_\theta$.
4:     Compute trajectory-level rewards $R_i$ and advantages $A^{(i)}$ for each $\tau^{(i)} \in \mathcal{B}$.
5:     **for** each successful trajectory $\tau^{(i)}$ with $R_i > 0$ **do**
6:       **for** each step $t$ in trajectory $\tau^{(i)}$ **do**
7:         Construct cognitive group $e_t^{(i)}$ by generating thinking processes under all 4 levels.
8:         Compute action prediction confidence $C_t^{(i,k)}$ for each level $k$
9:         Normalize confidence scores $C_{\text{norm},t}^{(i,k)}$ within the cognitive group using Eq. 2.
10:        Compute confidence-aware weights $g(C_t^{(i,k)})$ using Eq. 3.
11:        Compute step-level advantages $A_t^{(i,k)}$ using Eq. 4.
12:       **end for**
13:     **end for**
14:     For failed trajectories ($R_i \leq 0$), keep original advantages without cognitive group expansion.
15:     Update policy $\pi_\theta$ by maximizing the CoPO objective (Eq. 5).
16: **end for**
17: **Output:** Optimized policy $\pi_\theta$.

---

# E. Implementation Details

## E.1. Training Data Collection

We construct training datasets for both CoSFT and CoPO using the training splits of ALFWorld (2,420 simulations) and ScienceWorld (2,120 simulations). For CoSFT, we randomly sample 500 environments from each benchmark to create a dataset with uniform coverage across the four cognitive levels. For each sampled environment, we first collect expert trajectories using GPT-4o without reasoning augmentation, then prompt GPT-4o to generate structured thinking processes based on a randomly selected cognitive level. Prompt templates for different levels are shown in Listing 4. This procedure yields roughly 25% of data for each level, enabling the model to learn consistent cognitive patterns across all thinking modes. An example of the final data format is shown as follow.

*Listing 3.* Example of CoSFT training data format showing the structured output with cognitive level, thinking process, and action.

```
CoSFT Example:
{"conversations":
[{"from": "human", "value": "You are an agent for science world. Every round I will give
you an observation, you have to respond an action based on the observation to finish the
given task.
Here are the actions you may take: [valid actions omitted]
There are four thinking levels:
Level 1 - Instinctive Response: Immediate reaction based on intuition, no analysis.
Level 2 - Situational Awareness: Assess current state and available actions before acting.
Level 3 - Experience Integration: Reflect on past actions and outcomes to inform current
decisions.
Level 4 - Strategic Planning: Assess the task goal, past lessons, and current state to
analyze the future impact of each candidate action and optimize the decision.
At each step, you must first choose an appropriate level of thinking (one of the four
levels) to respond based on the given scenario. The chosen level MUST be enclosed within <
level> </level> tags.
Next, reason step-by-step using the chosen thinking level. This reasoning process MUST be
enclosed within <think> </think> tags. For Level 1 (Instinctive Response), use the fixed
text: "Okay, I think I have finished thinking." For Levels 2-4, provide detailed reasoning
 as shown in examples. Once you've finished your reasoning, you should choose an
admissible action for current step and present it within <action> </action> tags.
[Output format examples omitted]
```

```
Your current task is: Your task is to boil water. For compounds without a boiling point,
combusting the substance is also acceptable. First, focus on the substance. Then, take
actions that will cause it to change its state of matter. Your current observation is:
Your task is to boil water. For compounds without a boiling point, combusting the
substance is also acceptable. First, focus on the substance. Then, take actions that will
cause it to change its state of matter.This room is called the hallway. In it, you see: a
picture, a substance called air, the agent. You also see: A door to the green house (that
is open), A door to the living room (that is open), A door to the art studio (that is open
), A door to the kitchen (that is open), A door to the bedroom (that is open), A door to
the workshop (that is open). Now it's your turn to generate next step response."},
{"from": "gpt", "value": "<level>1</level><think>Okay, I think I have finished thinking.</
think><action>open door to kitchen</action>?}
],
"system": ""}
```

For CoPO, we use the remaining environments: 1,920 from ALFWorld and 1,620 from ScienceWorld. We adopt an online learning setup where the agent collects trajectories through environment interaction. During the cognitive group expansion process (see Section 3.4), we apply the same prompt templates (Listing 4) to generate thinking processes at all four levels while keeping the actions fixed.

*Listing 4.* Prompt templates for generating thinking processes at different cognitive levels.

```
Cognitive Level 2
Based on the following information, generate a thinking process that leads to the next
step action.

Task Description: {task_description}
Observation: {current_obs}
History of Previous Actions and Observations:
{history}
Next Action: {action}

You need to generate a thinking process following the exact format:
<think>
Current state: [Analyze the current environment state]
Available actions: [What actions are possible right now]
Reasoning: [Choose the best action and explain why]
</think>

Cognitive Level 3
Based on the following information, generate a thinking process that leads to the next
step action.

Task Description: {task_description}
Observation: {current_obs}
History of Previous Actions and Observations:
{history}
Next Action: {action}

You need to generate a thinking process following the exact format:
<think>
Goal: [What needs to be accomplished]
Current state: [Analyze the current environment state]
Available actions: [What actions are possible right now]
Reflection: [How effective were recent actions, what was learned]
Reasoning: [Choose the best action based on experience]
</think>

Cognitive Level 4
Based on the following information, generate a thinking process that leads to the next
step action.

Task Description: {task_description}
Observation: {current_obs}
```

```
History of Previous Actions and Observations:
{history}
Next Action: {action}

You need to generate a thinking process following the exact format:
<think>
Goal: [What needs to be accomplished]
Current state: [Analyze the current environment state]
Available actions: [What actions are possible right now]
Reflection: [How effective were recent actions, what was learned]
Evaluation: [Assess the potential effectiveness of each candidate action]
Reasoning: [Choose the optimal action with strategic reasoning]
</think>
```

### E.2. Training Details

Our COPO algorithm extends group-based RL frameworks by introducing *cognitive group expansion*, which enables step-wise credit assignment across different cognitive levels. Unlike GRPO, which applies trajectory-level updates, COPO generates alternate thinking processes for each decision point in successful trajectories (reward $R_i > 0$), keeping the observation $o_t$ and action $a_t$ fixed. This allows the model to learn which cognitive level best suits each state-action pair, based on observed outcomes. To reduce computational cost, we skip cognitive group expansion when all trajectories in a group share the same outcome reward (*i.e.*, all succeed or all fail). In such cases, the trajectory-level advantage is zero, so reweighting would yield no gradient signal regardless of the expansion.

Table 4 presents the training time over 150 iterations and the final performance of each online method (*i.e.*, GRPO, GiGPO, and COPO). Compared to GRPO, COPO requires approximately $1.4$–$1.5\times$ training time due to the additional expansion of the cognitive group. However, this remains acceptable given the substantial performance gains, demonstrating a favorable trade-off between computational cost and task performance.

*Table 4.* Training time and performance comparison of different methods on ALFWorld and ScienceWorld.

| Method | ALFWorld | | ScienceWorld | |
|---|---|---|---|---|
| | Training time | Performance | Training time | Performance |
| *Qwen2.5-7B-Instruct* | | | | |
| GRPO | 38h 25m 11s | 83.5 | 61h 2m 16s | 53.0 |
| GiGPO | 49h 33m 16s | 88.0 | 78h 21m 20s | 47.0 |
| CoPO | 54h 58m 27s | 92.5 | 86h 15m 50s | 72.0 |
| *Llama3.1-8B-Instruct* | | | | |
| GRPO | 36h 44m 38s | 84.0 | 51h 16m 3s | 50.5 |
| GiGPO | 54h 15m 59s | 89.5 | 88h 14m 38s | 64.0 |
| CoPO | 48h 24m 30s | 91.5 | 81h 36m 45s | 70.5 |

# F. Extended Results

## F.1. Cognitive Level Distribution and Training Dynamics

Section 4.2 presents results based on Qwen2.5-7B. Here, we extend the analysis using Llama3.1-8B to evaluate the generalizability of COPO. As shown in Figure 6, the cognitive collapse phenomenon appears consistently across both models. GRPO and GiGPO tend to converge to predominantly $\mathcal{L}_4$ thinking on both Qwen2.5-7B and Llama3.1-8B. In contrast, COPO maintains adaptive cognitive allocation throughout training. Figure 7 illustrates the evolution of cognitive level distribution during COPO training: the agent learns to allocate $\mathcal{L}_1$ for routine steps while preserving higher-level reasoning ($\mathcal{L}_2$-$\mathcal{L}_4$) for situations requiring deeper deliberation.

Figure 8 further shows that COPO converges faster across both architectures. On ALFWorld, it reaches over 90% success within 100 iterations, whereas GRPO and GiGPO plateau at lower performance despite extended training. On ScienceWorld, COPO exceeds a 70% success rate, while baselines remain below 65%. These results suggest that confidence-aware, step-level credit assignment in COPO offers more effective learning signals than trajectory-level methods, regardless of the underlying base model.

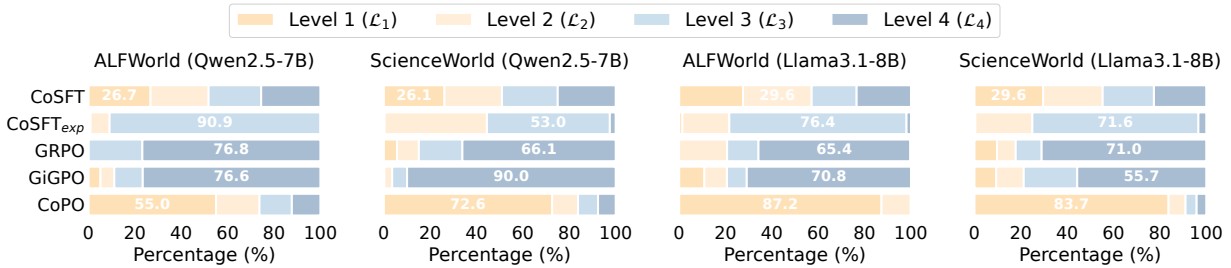

Figure 6. Cognitive level distribution after training. All RL methods (GRPO, GiGPO, COPO) are initialized from COSFT. While GRPO and GiGPO collapse to predominantly $\mathcal{L}_4$ thinking, COPO learns adaptive allocation.

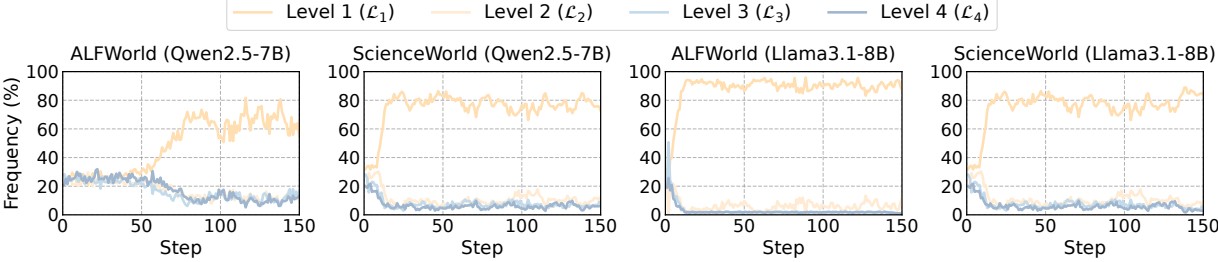

Figure 7. Dynamics of cognitive level distribution during training.

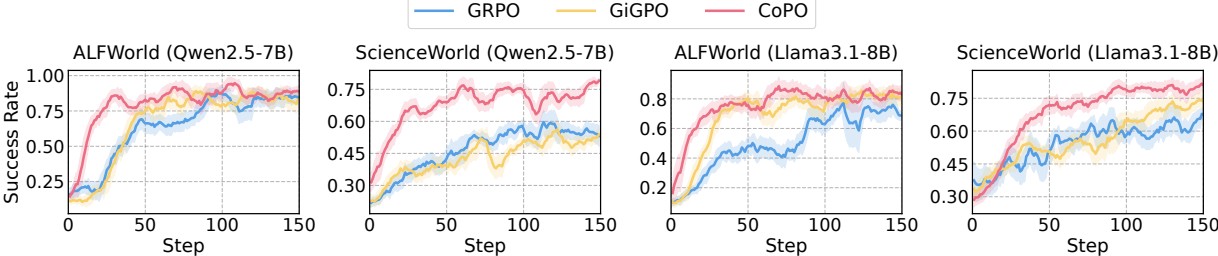

Figure 8. Training curves showing success rate across RL iterations for GRPO, GiGPO and COPO on ALFWorld and ScienceWorld. COPO achieves faster convergence to higher success rates.

## F.2. Additional Analysis on Cognitive Level Collapse

To further understand why baseline RL methods fail to maintain adaptive cognitive allocation, we compare the cognitive level distributions across trajectory progress for COPO, GiGPO, and GRPO on ScienceWorld, using Qwen2.5-7B as the base model. GRPO collapses to uniform $\mathcal{L}_4$ thinking due to coarse trajectory-level credit assignment. Although GiGPO

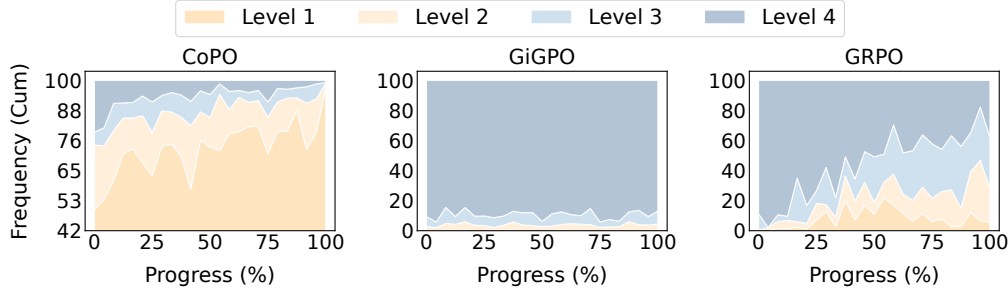

*Figure 9.* Cognitive level distributions across trajectory progress for CoPO, GiGPO, and GRPO on ScienceWorld. Both GRPO and GiGPO collapse to predominantly $\mathcal{L}_4$ thinking throughout trajectories, while CoPO exhibits stage-aware adaptation.

*Table 5.* Ablation studies on confidence metrics and training strategies. SR: Success Rate (%). #Tokens: Average tokens per trajectory. $\mathcal{L}_1$-$\mathcal{L}_4$: Cognitive level distribution (%).

| Variant | ALFWorld | | | | | | ScienceWorld | | | | | |
|---|---|---|---|---|---|---|---|---|---|---|---|---|
| | SR | #Tokens | $\mathcal{L}_1$ | $\mathcal{L}_2$ | $\mathcal{L}_3$ | $\mathcal{L}_4$ | SR | #Tokens | $\mathcal{L}_1$ | $\mathcal{L}_2$ | $\mathcal{L}_3$ | $\mathcal{L}_4$ |
| **Confidence Metric Ablation** | | | | | | | | | | | | |
| CoPO (Ours) | **92.5** | 1739.4 | 55.0 | 18.7 | 14.0 | 12.3 | **72.0** | **1543.4** | 72.6 | 11.2 | 8.6 | 7.6 |
| w/ *Max Probs* | 89.0 | 1664.9 | 65.7 | 12.3 | 11.4 | 8.4 | 69.5 | 2167.5 | 68.3 | 15.2 | 9.5 | 7.1 |
| w/ *Min Probs* | 81.5 | 1227.7 | 92.7 | 0.7 | 2.6 | 4.0 | 67.5 | 2063.6 | 69.5 | 12.7 | 10.7 | 7.1 |
| w/ *Entropy* | 79.0 | 1584.6 | 73.9 | 11.5 | 6.3 | 8.3 | 70.0 | 1873.9 | 72.0 | 12.7 | 8.6 | 6.7 |
| **Training Strategy Ablation** | | | | | | | | | | | | |
| w/o *Cold Start* | 86.0 | 2496.2 | 0.3 | 35.0 | 33.5 | 31.2 | 64.5 | 3750.6 | 0.0 | 27.1 | 35.5 | 37.3 |
| w/ $\text{CoSFT}_{\text{expert}}$ | 87.5 | 2871.6 | 0.0 | 7.9 | 92.1 | 0.0 | 62.0 | 4383.0 | 0.0 | 24.3 | 69.1 | 0.6 |
| w/ *Corr & Incorr* | 83.0 | 1375.8 | 83.1 | 4.3 | 7.7 | 4.8 | 66.0 | 1755.6 | 77.6 | 9.9 | 6.9 | 5.6 |

introduces step-level grouping, it similarly collapses, maintaining over 90% $\mathcal{L}_4$ usage throughout the trajectory with minimal stage-wise variation (Figure 9). This collapse arises from biased credit assignment: in GiGPO's state-based grouping, steps with deeper thinking often appear in successful trajectories and receive higher advantages, regardless of whether such depth was actually necessary. As a result, GiGPO defaults to deeper thinking as a universally safe choice, rather than allocating cognitive effort adaptively. In contrast, CoPO employs confidence-aware reweighting to assess the appropriateness of each cognitive level at every step. It estimates how confidently the model predicts actions under different thinking modes, providing a fine-grained signal that guides more adaptive cognitive allocation.

### F.3. Ablation Study Results

The ablation studies in Section 4.4 use Qwen2.5-7B. To assess generalizability, we conduct the same experiments on Llama3.1-8B. As shown in Table 5, we observe consistent trends: average log-probability remains the most effective confidence metric; balanced CoSFT initialization is critical to prevent cognitive level collapse; and restricting reweighting to successful trajectories mitigates adverse incentives that otherwise lead to $\mathcal{L}_1$ dominance.

## G. Case Study

We analyze a representative trajectory from ScienceWorld's *use thermometer* task using Qwen2.5-7B as the base model (Figure 10). At initialization (steps 1), the agent employs $\mathcal{L}_4$ (Strategic Planning) to formulate a global plan: locate the thermometer, identify the measurement target, and execute the measurement. During the execution phase (steps 4-12), the agent predominantly uses $\mathcal{L}_1$ (Instinctive Response) for routine navigation and object manipulation. When encountering an unexpected observation at step 8 ("the door is locked"), the agent switches to $\mathcal{L}_3$ (Experience Integration) to reflect on the obstacle and adjust the plan. This adaptive cognitive switching demonstrates that COGROUTER learns to allocate cognitive depth based on situational demands, achieving task success with 62% fewer tokens than GRPO's uniform $\mathcal{L}_4$ thinking.

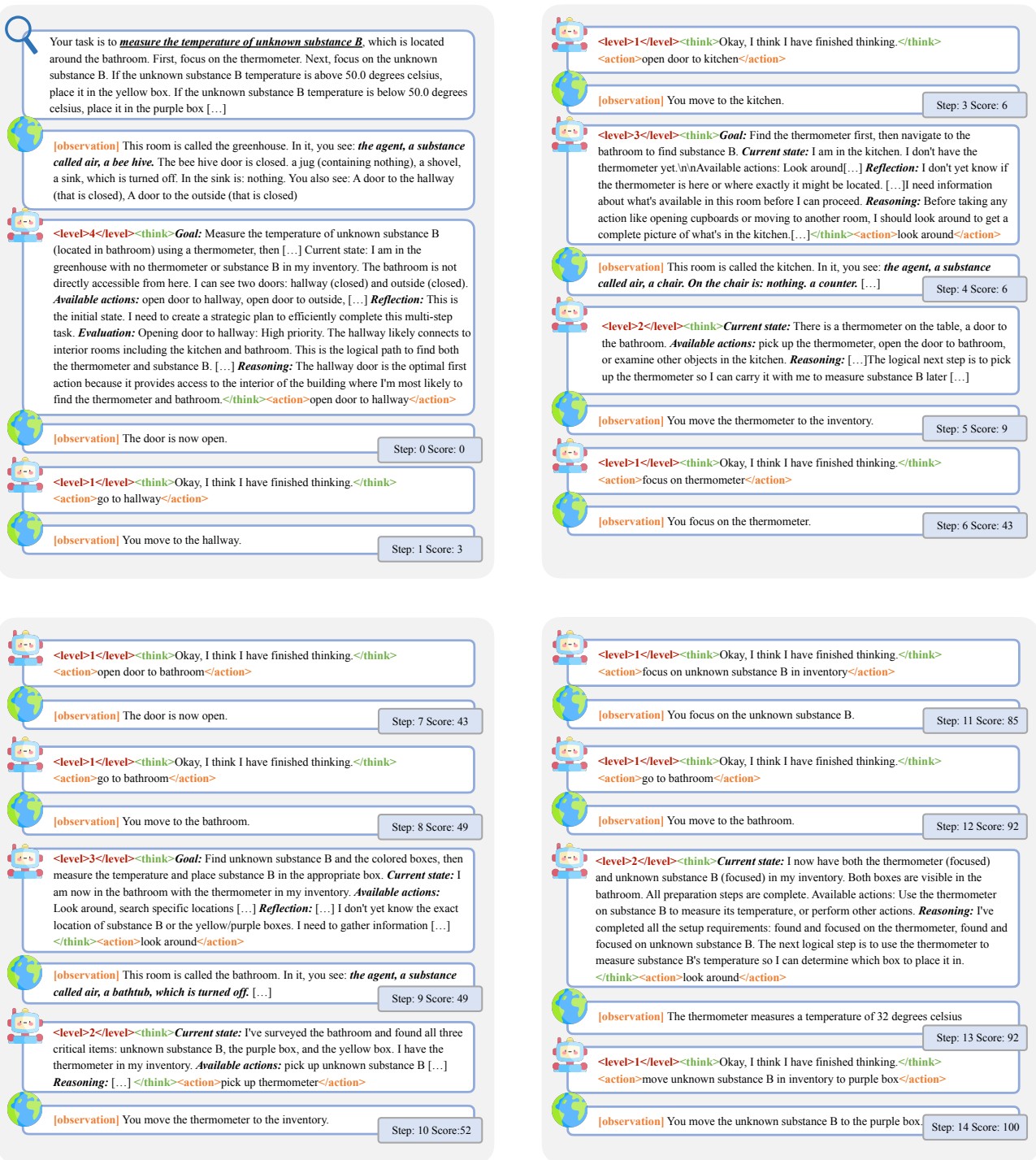

*Figure 10.* Case study of COGROUTER's adaptive cognitive switching on ScienceWorld's *use thermometer* task. The agent adaptively switches cognitive levels based on situational demands: employing $\mathcal{L}_4$ for initial strategic planning (step 0), $\mathcal{L}_1$ for routine navigation and manipulation (steps 6-8), $\mathcal{L}_3$ for obstacle reflection when encountering unexpected situations (step 9).

# H. Baselines

## H.1. Prompt-based Methods

**ReAct** (Yao et al., 2023b) is a prompting framework that interleaves thinking processes with task-specific actions. For both ALFWorld and ScienceWorld, we prompt the model to generate a thought before each action using the format: *Thought: ..., Action: ....* This format is applied to both frontier models (GPT-4o, DeepSeek-V3, Claude-4-Sonnet, OpenAI-o3, DeepSeek-R1, Gemini-2.5-Pro) and fine-tuned open-source models (Qwen2.5-7B-Instruct, Llama3.1-8B-Instruct). **Reflexion** (Shinn et al., 2023) is a self-refinement framework that enables agents to learn from trial-and-error. After each failed episode, the agent generates a reflection summarizing the failure and proposing improvements, which is prepended to the context in subsequent trials for the same task. We allow a maximum of 3 trials per task.

## H.2. Training-based Methods

We consider both offline and online methods using Qwen2.5-7B and Llama3.1-8B as base models. Detailed training settings are provided in Table 6.

**Offline Methods.** For **SFT**, we construct a supervised dataset by randomly sampling 500 environments from the ALFWorld and ScienceWorld training sets, consistent with CoSFT. For each environment, we collect expert trajectories and prompt GPT-4o to generate the corresponding reasoning steps in ReAct format, producing ReAct-style successful trajectories. These are used to fine-tune the base model via standard language modeling loss. **ETO** (Song et al., 2024) employs Direct Preference Optimization (DPO) (Rafailov et al., 2024), training on contrastive trajectory pairs. Preferred examples are taken from the SFT-generated successful trajectories, while rejected ones are sampled by rolling out the base model in the same environments.

**Online Methods.** All online RL methods are trained for 150 iterations on environments not used in SFT: 1920 in ALFWorld and 1620 in ScienceWorld. In each iteration, we randomly sample 128 environments to collect trajectories for policy updates. All models are initialized from the same CoSFT cold-start checkpoint for fair comparison. **GRPO** (Shao et al., 2024) performs trajectory-level policy updates by grouping trajectories into 8 clusters. **GiGPO** (Feng et al., 2025) extends GRPO with step-level credit assignment using anchor state grouping. We use the same group size (8) for computing trajectory-level advantages. For step-level rewards, we set the weighting coefficient $\omega = 1.0$ and the discount factor $\gamma = 0.95$. **AdaptThink** (Zhang et al., 2025a) teaches models to adaptively select between thinking and non-thinking modes. We adapt it to our multi-step agentic scenarios. We first perform SFT cold start by sampling 500 environments and constructing a balanced 1:1 dataset of thinking and non-thinking steps, where non-thinking uses `<think></think>` (empty think tags) to skip thinking. We then train with GRPO using a modified reward $r(\tau) = \frac{1}{|\tau|} \sum_{t=1}^{|\tau|} \mathbf{1}(\text{NoThink}_t) \cdot \delta + R(\tau)$, where $\mathbf{1}(\text{NoThink}_t)$ indicates whether step $t$ uses non-thinking mode (*i.e.*, generates `<think></think>`), and $R(\tau)$ is the task reward from §3.4. The first term represents the proportion of non-thinking steps in each trajectory. We set $\delta = 0.05$ and use importance sampling during training to balance exploration of both modes, with all other settings following GRPO.

# I. Hyperparameters

Full hyperparameter details are provided in Table 6. For offline methods, we use the LLaMA-Factory codebase.[4] For online methods, we implement our algorithms based on the VERL framework (Sheng et al., 2025).

*Table 6.* Hyperparameters for all experiments across different methods and benchmarks. All RL methods (GRPO, GiGPO, CoPO) use the same hyperparameters for fair comparison.

| | | Qwen2.5-7B | | Llama3.1-8B | |
|---|---|---|---|---|---|
| | | ALFWorld | ScienceWorld | ALFWorld | ScienceWorld |
| SFT | Learning Rate | 2e-6 | 2e-6 | 2e-6 | 2e-6 |
| | Batch Size | 32 | 32 | 32 | 32 |
| | Number of Epoch | 3 | 3 | 3 | 3 |
| | Max Sequence Length | 8192 | 8192 | 8192 | 8192 |
| CoSFT (CoSFT$_{exp}$) | Learning Rate | 2e-6 | 2e-6 | 2e-6 | 2e-6 |
| | Batch Size | 32 | 32 | 32 | 32 |
| | Number of Epoch | 3 | 3 | 3 | 3 |
| | Max Sequence Length | 8192 | 8192 | 8192 | 8192 |
| ETO | Learning Rate | 2e-6 | 2e-6 | 2e-6 | 2e-6 |
| | Batch Size | 32 | 32 | 32 | 32 |
| | Number of Epoch | 3 | 3 | 3 | 3 |
| | Max Sequence Length | 8192 | 8192 | 8192 | 8192 |
| GRPO | Max Prompt Length | 18000 | 30000 | 18000 | 30000 |
| | Max Response Length | 1024 | 1024 | 1024 | 1024 |
| | Group Size | 8 | 8 | 8 | 8 |
| | Groups per Rollout | 16 | 16 | 16 | 16 |
| | KL Coefficient | 0.1 | 0.2 | 0.1 | 0.2 |
| | Learning Rate | 5e-7 | 5e-7 | 2e-7 | 2e-7 |
| | Rollout Temperature | 1.0 | 1.0 | 1.0 | 1.0 |
| | Validation Temperature | 0.4 | 0.4 | 0.4 | 0.4 |
| | Mini-batch Size | 64 | 64 | 64 | 64 |
| | Iteration | 150 | 150 | 150 | 150 |
| GiGPO | Max Prompt Length | 18000 | 30000 | 18000 | 30000 |
| | Max Response Length | 1024 | 1024 | 1024 | 1024 |
| | Group Size | 8 | 8 | 8 | 8 |
| | Groups per Rollout | 16 | 16 | 16 | 16 |
| | KL Coefficient | 0.1 | 0.2 | 0.1 | 0.2 |
| | Learning Rate | 5e-7 | 5e-7 | 2e-7 | 2e-7 |
| | Rollout Temperature | 1.0 | 1.0 | 1.0 | 1.0 |
| | Validation Temperature | 0.4 | 0.4 | 0.4 | 0.4 |
| | Mini-batch Size | 64 | 64 | 64 | 64 |
| | Iteration | 150 | 150 | 150 | 150 |
| CoPO | Max Prompt Length | 18000 | 30000 | 18000 | 30000 |
| | Max Response Length | 1024 | 1024 | 1024 | 1024 |
| | Group Size | 8 | 8 | 8 | 8 |
| | Groups per Rollout | 16 | 16 | 16 | 16 |
| | KL Coefficient | 0.1 | 0.2 | 0.1 | 0.2 |
| | Learning Rate | 5e-7 | 5e-7 | 2e-7 | 2e-7 |
| | Rollout Temperature | 1.0 | 1.0 | 1.0 | 1.0 |
| | Validation Temperature | 0.4 | 0.4 | 0.4 | 0.4 |
| | Mini-batch Size | 64 | 64 | 64 | 64 |
| | Iteration | 150 | 150 | 150 | 150 |

---

[4]https://github.com/hiyouga/LLaMA-Factory

# J. Necessity of Adaptive Thinking

To evaluate the importance of adaptive thinking, we compare CogRouter against two types of baselines on ALFWorld using Qwen2.5-7B: *1)* **Fixed cognitive levels**, where separate models are trained via SFT using a single reasoning format ($\mathcal{L}_1$, $\mathcal{L}_2$, $\mathcal{L}_3$, or $\mathcal{L}_4$), followed by GRPO; and *2)* **FreeForm**, trained with standard SFT using an unstructured reasoning format, also followed by GRPO. As shown in Figure 11, CogRouter achieves a 92.5% SR with only 1739.42 tokens, outperforming all baselines in both accuracy and efficiency. The fixed-level models exhibit a clear trade-off: $\mathcal{L}_1$ achieves 76.5% SR with 357.04 tokens, while $\mathcal{L}_4$ reaches 86.5% SR at the cost of 4640.98 tokens, still 6% below CogRouter despite using 2.7× more tokens. FreeForm attains 81.5% SR with 4068.88 tokens, suggesting it applies deep reasoning uniformly, without adjusting to task complexity. These results indicate that neither shallow nor deep reasoning alone is sufficient: shallow approaches underperform on complex steps, while deep ones waste computation on simple cases. Adaptive thinking addresses this by dynamically adjusting reasoning depth to task demands.

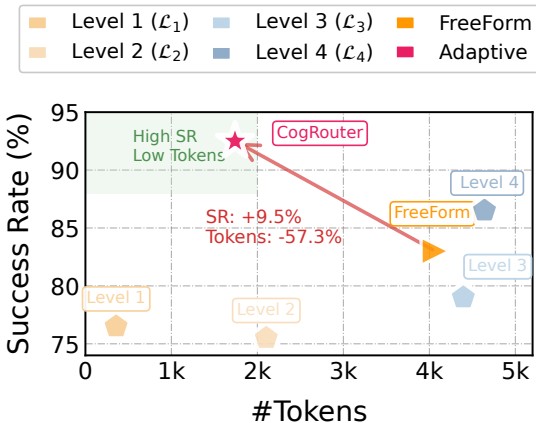

*Figure 11.* Comparison of cognitive configurations on ALFWorld (Qwen2.5-7B). CogRouter outperforms both fixed cognitive levels and unstructured reasoning, highlighting the need for adaptive thinking.

### J.1. Analysis of Contextual Cognitive-Level Selection

We further analyze how CogRouter selects cognitive levels under different environmental states. Specifically, for each cognitive level, we collect the states where the level receives the highest confidence during training and report the top-3 most frequent observations. The results are shown in Table 7.

| Level | Top Observations | Interpretation |
|---|---|---|
| L1 | "The door is now open" (40.2%); "The door is not open" (18.3%); "You focus on the [object]" (17.0%) | Simple environmental feedback requiring reflexive actions. |
| L2 | "You move to the [other room]" (34.9%); "You move to the hallway" (19.6%); "[circuit] anode/cathode connection state" (17.2%) | Arrival at new environments requiring situational awareness. |
| L3 | "No known action matches that input" (82.0%); "Connections must specify..." (12.8%); "The [container] isn't open, can't see inside" (2.8%) | Error signals requiring reflection and plan revision. |
| L4 | "(Task Start)" (57.7%); "This room is called the [room]..." (19.7%); "Inside the [container] is:..." (10.2%) | Task initialization and information-rich states requiring strategic planning. |

*Table 7.* Top observations associated with each cognitive level when the level receives the highest confidence during training. The results show that different levels are selected for semantically distinct state types, including routine feedback, situational assessment, error recovery, and strategic planning.

The results reveal clear contextual specialization across cognitive levels. $\mathcal{L}_1$ is mainly selected for routine environmental feedback, such as whether a door has been opened, where the next action can often be chosen directly. $\mathcal{L}_2$ is associated with entering new locations or observing structured state changes, where the agent needs to assess the current situation before acting. $\mathcal{L}_3$ is strongly associated with error states, such as invalid actions or failed preconditions, which require reflection over previous attempts and plan revision. $\mathcal{L}_4$ is most frequently selected at task initialization and information-rich states, where global planning and candidate-action evaluation are useful.

These findings show that the confidence-based routing signal induces meaningful state-dependent level allocation. The selected levels are not uniformly biased toward longer or more detailed reasoning. Instead, CogRouter assigns lightweight levels to routine transitions and deeper levels to states that require reflection or strategic planning. This behavior is consistent with the stage-wise routing patterns observed at test time in Figure 3, further demonstrating that CogRouter learns to adapt cognitive depth according to contextual demands.

