# OpenReview forum: "Think Fast and Slow: Step-Level Cognitive Depth Adaptation for LLM Agents"
_ICML.cc/2026/Conference — ICML 2026 regular_

### Official Review · Reviewer_n5Jd · 2026-03-02

**Soundness:** 3
**Presentation:** 3
**Significance:** 3
**Originality:** 3
**Overall Recommendation:** 4
**Confidence:** 4

**Summary:**

This paper introduces CogRouter, a framework for training large language model (LLM) agents to dynamically adapt their cognitive depth at each step of multi-turn decision-making tasks. Grounded in Adaptive Control of Thought-Rational (ACT-R) theory, the framework defines four hierarchical cognitive levels (from instinctive to strategic reasoning). CogRouter uses a two-stage training pipeline: Cognition-aware Supervised Fine-tuning (CoSFT) to instill level-specific patterns with balanced data, and Cognition-aware Policy Optimization (CoPO), a reinforcement learning algorithm that performs step-level credit assignment via confidence-aware advantage reweighting. The key insight is that the appropriate cognitive depth should maximize action confidence. Experiments on ALFWorld and ScienceWorld benchmarks demonstrate that CogRouter achieves state-of-the-art performance, significantly outperforming major baselines while achieving substantial efficiency gains by reducing token usage.

**Compliance With Llm Reviewing Policy:**

Affirmed.

**Final Justification:**

The authors’ rebuttal addressed most of my concerns and strengthened the paper. In particular, the additional results on resource tradeoffs, broader environments, and unseen-task generalization improve both the empirical scope and practical value of the work. Although I still think the discussion of cognitive granularity could be stronger, the new evidence makes the contribution more convincing overall and reinforces my positive assessment.

**Key Questions For Authors:**

1. **How much does discrete cognitive-level granularity limit** the agent? Would continuous or learnable granularity improve adaptation, or is there a cognitive/empirical justification for sticking with four pre-defined levels?
2. **What are the computational tradeoffs** in terms of training and inference cost, especially for scaling to much larger models or real-time deployment settings?
3. **What is the method’s robustness to open-ended or real-world environments** not seen during training? Can it generalize to, for example, new ALFWorld/ScienceWorld tasks or to entirely different benchmarks/environments?

**Limitations:**

While the method shows strong performance and efficiency on ALFWorld and ScienceWorld, the evaluation is still limited in scope, and the approach incurs moderately higher training cost with a fixed four-level discrete depth design. Additional studies on more diverse environments, along with a clearer compute–benefit analysis and exploration of finer-grained (or continuous) depth control, would help strengthen the generality and practical guidance of the conclusions.

**Strengths And Weaknesses:**

### Strengths

1. **Well-motivated hierarchical cognitive adaptation grounded in ACT-R**
   The paper’s explicit *step-level* adaptive selection among **four cognitive depths**, motivated by cognitive science (ACT-R), is clearly articulated and meaningfully advances prior “binary-switch” adaptive reasoning approaches. The overall architecture (Figure 2) convincingly operationalizes the cognitive theory into a concrete training and inference mechanism.

2. **Strong state-of-the-art performance with substantial efficiency gains**
   The main results (Table 1) show CoPO achieving the best performance on **both ALFWorld and ScienceWorld**, while also delivering large token-efficiency improvements (reported up to **62% fewer tokens** than strong baselines). The gains appear robust across multiple backbone models, supporting the method’s generality.

3. **Thorough experimental validation and analysis**
   The paper includes extensive experiments, ablations, and diagnostics (Tables 1, 2, 4, 5; Figures 3–6, 10–11), covering both quantitative outcomes and qualitative allocation behaviors. In particular, the visual analyses (e.g., cognitive level distributions and allocation dynamics) strengthen the credibility of the core claim that CoPO learns meaningful, context-dependent depth allocation.

---

### Weaknesses

1. **Training scalability and computational overhead are under-discussed**
   Table 4 reports CoPO requiring roughly **1.4–1.5× longer training time** than GRPO/GiGPO. While the accuracy and efficiency gains are compelling, the paper would benefit from a clearer discussion of the *resource–performance tradeoff* (e.g., wall-clock cost vs. downstream token savings), and possibly additional scaling analysis (larger environments, longer horizons, or more expensive backbones) to help practitioners assess deployment feasibility.

2. **Discrete four-level cognitive granularity may be limiting**
   Restricting cognitive depth to **exactly four discrete levels** (per ACT-R) may oversimplify the continuum of reasoning demands in more complex or unstructured settings. It would be valuable to evaluate whether **more granular** (or continuous) depth control improves performance, especially given related directions such as **Dynamic Nested Depth (DND)**:
   *DND: Boosting Large Language Models with Dynamic Nested Depth.*

3. **Limited environment diversity constrains empirical scope**
   Although ALFWorld and ScienceWorld are strong embodied testbeds, the evaluation does not include additional settings such as other embodied simulators, web-agent benchmarks, or real-world task datasets. This limits how broadly one can generalize the empirical claims, especially regarding robustness to different interaction styles, observation spaces, or long-horizon credit assignment patterns.

---

> ### Author Rebuttal · Authors · 2026-03-30
>
> > **W1 & Q2**: Training scalability and computational overhead
>
> We appreciate this concern and provide the resource–performance analysis as follows:
>
> | Method || ALFWorld |||| ScienceWorld |||
> |--------|----------|-----|--------|------|----------|-----|--------|------|
> | | Training | SR | #Tokens | Inference | Training | SR | #Tokens | Inference |
> | GRPO | 38h 25m 11s | 83.5 | 4994.6 | 282.26s | 61h 2m 16s | 53.0 | 3739.9 | 279.93s |
> | GiGPO | 49h 33m 16s | 88.0 | 2955.5 | 281.66s | 78h 21m 20s | 47.0 | 4602.8 | 293.60s |
> | **CoPO** | **54h 58m 27s** | **92.5** | **1739.4** | **179.84s** | **86h 15m 50s** | **72.0** | **1543.4** | **153.24s** |
>
> **Resource–performance tradeoff:** CoPO incurs ~1.4× training time versus GRPO, but yields: (1) +9-19% SR improvement, (2) 60-65% token reduction, and (3) **36-45% faster inference**. We will include these results in the revised manuscript and plan to validate on larger frontier models in future work.
>
> > **W2 & Q1**: Discrete four-level cognitive granularity
>
> We appreciate this insightful question. We address it as follows:
>
> - **Theoretical grounding.** Our 4-level hierarchy is grounded in ACT-R's four qualitatively distinct cognitive stages, from automatic procedural execution to deliberate declarative reasoning (Section 3.2, Lines 138-181). This provides a principled design rather than an arbitrary choice.
>
> - **Comparison with Continuous reasoning depth.** We compare against FreeForm, which allows continuous reasoning depth without discrete level selection or structured templates, on ALFWorld with Qwen2.5-7B (Appendix J):
>
> | Configuration | SR | #Tokens |
> |---------------|-----|---------|
> | **4-Level (Ours)** | **92.5** | 1739.4 |
> | FreeForm | 81.5 | 4068.9 |
>
> FreeForm allows arbitrary reasoning depth but lacks explicit cognitive anchors, making it difficult for the model to learn when to stop thinking. This results in verbose, unstructured outputs with lower performance. The 4-level hierarchy provides explicit anchors that enable learnable and efficient depth allocation.
>
> - **Token-level approaches.** Finer-grained methods like DND [1] target a **different objective**: training a token-level router to identify "hard" tokens for additional computation, improving prediction accuracy but not reducing token usage. In contrast, CoPO learns step-level routing for both task success and token efficiency. Moreover, token-level methods face challenges in agentic settings: supervision is sparse (only final rewards), and agentic tasks involve dynamic state spaces where each step receives new observations. SFT on limited correct trajectories cannot cover the diverse states encountered during inference. We will add discussion of this in the related work.
>
> [1] DND: Boosting LLMs with Dynamic Nested Depth. ICLR 2026.
>
> > **W3**: Environment diversity
>
> Thank you for this suggestion. We conduct experiments on **WebShop**, a realistic e-commerce benchmark. Using Qwen2.5-7B, we sample 500 of 6910 AgentGym [1] simulations for CoSFT and train on the remainder for 60 RL iterations. Evaluation on 200 held-out simulations:
>
> | Method | Score | SR | #Tokens |
> |--------|-------|-----|---------|
> | Qwen2.5-7B (ReAct) | 46.2 | 19.5 | 1584.3 |
> | GRPO (60 iter) | 70.2 | 55.5 | 2917.4 |
> | **CoPO (60 iter)** | **75.4** | **62.0** | **1723.6** |
>
> CoPO outperforms GRPO by +5.2 score and +6.5% SR while using **40% fewer tokens**. We will include full 150 iteration results in the main results of the revised manuscript.
>
> [1] AgentGym: Evolving Large Language Model-based Agents across Diverse Environments
>
> > **Q3**: Generalization to unseen tasks
>
> Our main experiments use the standard **L1 split** (test and train share the same task categories but differ in variants), consistent with prior work. To validate OOD generalization, we follow RLVMR [1] and evaluate on the **L2 split** (entire task categories are held out from training), the most challenging setting. Results with Qwen2.5-7B and Llama3.1-8B are as follows:
>
> | Method | ALFWorld || ScienceWorld ||
> |--------|----------|---------|----------|---------|
> | | SR | #Tokens | SR | #Tokens |
> | *Qwen2.5-7B* |
> | GRPO | 52.5 | 4486.3 | 26.5 | 3915.8 |
> | **CoPO** | **74.0** | **1924.7** | **32.0** | **1738.6** |
> | *Llama3.1-8B* |
> | GRPO | 45.5 | 4328.5 | 25.0 | 4512.3 |
> | **CoPO** | **77.5** | **1286.4** | **38.5** | **1352.8** |
>
> CoPO maintains strong advantages on L2: **+21.5% SR** (ALFWorld) and **+5.5% SR** (ScienceWorld) over GRPO with Qwen2.5-7B, while using **~57% fewer  fewer tokens**. We will include these results in revised manuscript.
>
> [1] RLVMR: REINFORCEMENT LEARNING WITH VERIFIABLE META-REASONING REWARDS FOR ROBUST
> LONG-HORIZON AGENTS, ICLR 2026.

---

> > ### Author Rebuttal · Reviewer_n5Jd · 2026-04-04
> >
> > The authors’ rebuttal has addressed most of my previous concerns, and I appreciate the clarifications provided. However, the response regarding the four-level cognitive granularity does not sufficiently resolve my concerns or answer the key questions raised. As this issue is central to the contribution, I am inclined to maintain my original score.

---

> > > ### Author Response · Authors · 2026-04-06
> > >
> > > We thank the reviewer for the follow-up. We take this concern seriously and provide new empirical evidence and theoretical justification.
> > >
> > > > **Evaluating different cognitive granularities**
> > >
> > > To systematically evaluate the impact of cognitive granularity, we compare configurations spanning from coarse to fine on ScienceWorld with Qwen2.5-7B. For the 2-level configuration, we retain only L1 (instinctive response) and L4 (strategic planning). For the 5-level configuration, we introduce L5 (Meta-reasoning), following the definition in RLVMR [1]: the agent tracks progress against the overall plan and ensures actions remain aligned with task goals. We conduct experiments on ScienceWorld with Qwen2.5-7B for 30 CoPO iterations:
> > >
> > > > | Configuration | Score | #Tokens | Training Time |
> > > > |---|---|---|---|
> > > > | CoPO 2-level (L1 + L4) | 43.88 | 1450.5 | 9h 22m |
> > > > | CoPO 4-level (Ours) | 59.16 | 1608.3 | 14h 16m |
> > > > | CoPO 5-level (L1-L5) | 58.74 | 1754.7 | 16h 43m |
> > >
> > > The results reveal that 4 levels represent an effective balance, consistent with both ACT-R and HCCT [2] which independently converge on a four-level hierarchy for cognitive control:
> > >
> > > - **Too few levels lack coverage:** Reducing to 2 levels significantly hurts performance (43.88 vs. 59.16), as it misses intermediate cognitive demands such as situational awareness (L2) and reflection (L3).
> > > - **Too many levels lack differentiation:** Extending to 5 levels yields no improvement but increases token usage (+9%) and training time (+17%). L4's strategic planning inherently incorporates progress tracking and self-evaluation, leaving limited room for L5 to add value. Additionally, **more levels increase the difficulty for the model to select the appropriate level during routing.**
> > >
> > > We will include full 150 iteration results in the main results of the revised manuscript.
> > >
> > > Importantly, our framework is highly flexible in both the number and content of cognitive levels: CoPO's reweighting mechanism generalizes to any configuration, and adding or redefining levels only requires modifying the prompt template (Appendix C). **While finer granularity is architecturally supported, our results indicate that 4 levels already achieve optimal performance for the current benchmarks.**
> > >
> > > [1] RLVMR: Reinforcement Learning with Verifiable Meta-Reasoning Rewards for Robust Long-Horizon Agents. ICLR 2026.
> > >
> > > [2] An information theoretical approach to prefrontal executive function. Trends in Cognitive Sciences, 2007.

---

### Official Review · Reviewer_Yz4c · 2026-03-09

**Soundness:** 3
**Presentation:** 3
**Significance:** 3
**Originality:** 3
**Overall Recommendation:** 4
**Confidence:** 3

**Summary:**

The paper introduces COGROUTER, a novel framework designed to address cognitive rigidity in LLM agents by enabling them to dynamically adapt their reasoning depth at each interaction step. The authors employ a two-stage training pipeline consisting of Cognition-aware Supervised Fine-tuning (COSFT) to instill stable reasoning patterns and Cognition-aware Policy Optimization (COPO), which utilizes a novel reinforcement learning objective to perform step-level credit assignment based on action prediction confidence . Experimental results on the ALFWorld and ScienceWorld benchmarks demonstrate that the proposed method achieves good results.

**Compliance With Llm Reviewing Policy:**

Affirmed.

**Final Justification:**

Authors addressed most of my concerns. I keep my positive score.

**Key Questions For Authors:**

Please refer to Strengths And Weaknesses.

**Limitations:**

Yes

**Strengths And Weaknesses:**

## Strengths

1. This paper is well-written. The "cognitive rigidity" problem is clearly motivated.

2. The use of two distinct benchmarks and two different base models demonstrates robustness of the proposed method.

3. This work addresses a critical efficiency-performance trade-off in LLM agents and  significantly reduces token usage.

##  Weaknesses

1. Dependence on Expert Models: The COSFT stage relies on GPT-4o to generate "expert" thinking processes. This suggests the model's performance may be capped by the quality and biases of the teacher model.

2. Hierarchy Scalability: Why was a 4-level hierarchy chosen? Would like to see experiment with more or fewer levels (e.g., a binary switch vs. a 6-level hierarchy), and how does that impact the confidence-aware reweighting?

3. Scaling Concerns: While manageable for the 7B and 8B models tested, this additional generation step could become a significant bottleneck when training much larger frontier models.

---

> ### Author Rebuttal · Authors · 2026-03-30
>
> > **W1**: Dependence on Expert Models.
>
> We would like to clarify the role of GPT-4o in our framework. **GPT-4o is not used as a routing teacher**, it does not decide which cognitive level to use at each step. GPT-4o's roles are:
>
> 1. **Data collection:** Generate successful observation-action trajectories.
> 2. **Format completion:** Given a *randomly sampled* cognitive level, fill in the corresponding thinking format.
>
> As mentioned in Lines 189-194, the cognitive level at each step is **randomly assigned** when constructing CoSFT data (not selected by GPT-4o), ensuring balanced distribution across all four levels. We initialize RL from CoSFT (balanced distribution) rather than CoSFT$\_{exp}$ (expert-selected). Table 2 ablation confirms that CoSFT$\_{exp}$ leads to biased collapse (69.1% L3 dominance) and degraded performance (SR drops from 72.0% to 62.0%).
>
> Furthermore, the RL stage is **fully on-policy**: CoPO learns cognitive routing through environment interaction, with no dependence on GPT-4o's routing decisions. As shown in Table 1, Qwen2.5-7B after CoPO achieves 92.5% SR on ALFWorld and 72.0% SR on ScienceWorld, outperforming GPT-4o (61.5% and 22.5%) by 30-50%. This confirms that CoPO is not capped by the teacher model.
>
> > **W2**: Hierarchy Scalability
>
> Thank you for this insightful question. Our 4-level design is grounded in **ACT-R theory** [1]. ACT-R models human cognition as a spectrum from automatic procedural execution to deliberate declarative reasoning, with four qualitatively distinct stages (Section 3.2, Lines 138-181):
>
> | ACT-R Stage | Cognitive Process | Our Level | Our Implementation |
> |-------------|-------------------|-----------|-------------------|
> | Automatic procedural | Compiled production rules execute | L1 | Immediate response with no explicit reasoning |
> | Goal-directed procedural | Situational information maintained in working memory | L2 | Assess Current State and Available Actions |
> | Knowledge compilation | Declarative memories retrieved and consolidated | L3 | Goal + Reflection on past actions |
> | Declarative reasoning | Chunk retrieval with strategic evaluation | L4 | Simulation + Evaluation of candidate actions |
>
> **Empirical comparison:** We have included AdaptThink (binary switch between Think vs. Non-Think) and FreeForm (no fixed cognitive levels) in Table 1 and Appendix J. Results using Qwen2.5-7B on ALFWorld:
>
> | Configuration | SR | #Tokens |
> |---------------|-----|---------|
> | Binary (AdaptThink) | 81.5 | 1372.1 |
> | **4-Level (Ours)** | **92.5** | 1739.4 |
> | FreeForm (no levels) | 81.5 | 4068.9 |
>
> Binary switching underperforms as it forces an all-or-nothing choice: either skip reasoning entirely (L1) or engage in full deliberation (L4). This misses intermediate cognitive demands where moderate reasoning (L2/L3) would suffice. FreeForm allows arbitrary reasoning depth but lacks explicit cognitive anchors, making it difficult for the model to learn when to terminate reasoning and resulting in verbose, unstructured outputs with lower performance. The 4-level hierarchy provides **explicit cognitive anchors** that enable learnable and efficient routing.
>
> [1] Anderson, J. R. How Can the Human Mind Occur in the Physical Universe?
>
> > **W3**: Scaling Concerns
>
> Thank you for this concern. During training, we have implemented optimizations to minimize the overhead of cognitive group expansion (Appendix E.2):
>
> 1. **Selective expansion:** We only expand cognitive groups for successful trajectories (reward > 0)
> 2. **Skip redundant computation:** When all trajectories in a group share the same reward, expansion is skipped entirely since trajectory-level advantage is zero
>
> The training and inference cost comparison for Qwen2.5-7B is as follows:
>
> | Method | ALFWorld |||| ScienceWorld ||||
> |--------|----------|-----|--------|------|----------|-----|--------|------|
> | | Training Time | SR | #Tokens | Inference | Training Time | SR | #Tokens | Inference |
> | GRPO | 38h 25m 11s | 83.5 | 4994.6 | 392.26s | 61h 2m 16s | 53.0 | 3739.9 | 279.93s |
> | GiGPO | 49h 33m 16s | 88.0 | 2955.5 | 281.66s | 78h 21m 20s | 47.0 | 4602.8 | 323.60s |
> | **CoPO** | **54h 58m 27s** | **92.5** | **1739.4** | **179.84s** | **86h 15m 50s** | **72.0** | **1543.4** | **153.24s** |
>
> As shown in the table, CoPO requires approximately **1.4× training time** compared to GRPO, but this modest overhead yields substantial gains: +9.0% SR on ALFWorld and +19.0% SR on ScienceWorld. Furthermore, CoPO achieves **36-48% faster inference** due to complexity-aware routing that favors lightweight L1 reasoning for routine steps rather than collapsing to expensive L4 thinking. Overall, CoPO incurs slightly higher training cost but offers significant inference speedup, a favorable trade-off for real-world deployment.
>
> We will include these results in the revised manuscript and plan to validate on larger frontier models in future work.

---

> > ### Author Rebuttal · Reviewer_Yz4c · 2026-04-01
> >
> > Thanks for the rebuttal. I keep my score.

---

### Official Review · Reviewer_PvCE · 2026-03-13

**Soundness:** 3
**Presentation:** 4
**Significance:** 3
**Originality:** 3
**Overall Recommendation:** 5
**Confidence:** 3

**Summary:**

This paper proposes CogRouter, a framework that enables LLM agents to adaptively adjust their reasoning depth at each step during multi-step interactive tasks. Inspired by Kahneman's dual-process theory and grounded in ACT-R cognitive architecture, the method defines four cognitive levels (L1: Intuitive Reaction, L2: Situational Awareness, L3: Experience Integration, L4: Strategic Planning) and trains the agent to select the appropriate level per step.

The framework consists of two stages: (1) CoSFT, which uses GPT-4o to generate training data covering all four cognitive levels, ensuring balanced initialization; and (2) CoPO, a cognition-aware policy optimization method that performs step-level credit assignment using the model's prediction confidence to determine which cognitive level best fits each step. A key empirical finding is "Cognitive Level Collapse" — standard RL methods (GRPO/GiGPO) degrade to almost exclusively using L4 (deepest) thinking, wasting tokens on simple steps. CoPO prevents this collapse and achieves strong results on ALFWorld (92.5%) and ScienceWorld (72.0%) with Qwen2.5-7B, outperforming GRPO by 9.0 and 19.0 points respectively.

**Compliance With Llm Reviewing Policy:**

Affirmed.

**Final Justification:**

After reviewing the rebuttal, I maintain my score of **Accept**.

The authors have thoroughly addressed all three of my weaknesses with clarifications, new baselines, and additional experiments on WebShop, and I appreciate the thorough engagement.

That said, I did not raise my score higher, as I still have reservations about the broader **significance** of the work. In particular, the reliance on proprietary models and the use of pre-defined cognitive levels limit the generalizability and practical impact of the approach. These are not concerns introduced by the rebuttal, but rather inherent limitations of the current work that prevent a higher score.

**Key Questions For Authors:**

See Weakness， W1 and W2.

**Limitations:**

The authors acknowledge the limited scope of environments (two benchmarks) and model scales (7B-8B). However, the two evaluation environments share similar task structures, which is underaddressed as a limitation.

**Strengths And Weaknesses:**

### Strengths

1. **Well-motivated framework with novel empirical finding.** The "think on demand" idea — matching reasoning depth to step complexity — is intuitive and well-grounded in cognitive science (Kahneman's dual-process theory, ACT-R). Beyond the framework design, the paper identifies a new phenomenon, "Cognitive Level Collapse": standard RL methods (GRPO/GiGPO) collapse to predominantly L4 thinking even from balanced CoSFT initialization (Figure 3, Figure 6). The root cause analysis — trajectory-level credit assignment treats deep thinking as universally safe — is insightful.

2. **Effective technical solution.** CoPO's confidence-aware step-level credit assignment is elegant — using the model's own prediction confidence to evaluate cognitive level fitness at each step, then computing fine-grained advantage functions via temperature-scaled softmax normalization. This directly addresses the collapse problem.

3. **Strong empirical results with maintained cognitive diversity.** The performance gains are substantial (ALFWorld: +9.0 over GRPO; ScienceWorld: +19.0 over GRPO) with only ~1.4-1.5x training time overhead. CoPO preserves a balanced distribution across cognitive levels, with clear differentiation by task difficulty (Figure 5): short/simple tasks show dominant L1 usage, while case studies (Figure 10) show L4 for planning, L1 for navigation, and L3 for unexpected situations.

### Weaknesses

W1: **Lack of negative training signal.** CoPO only trains on successful trajectories via reject sampling — there is no negative gradient from failed trajectories. This setup differs from most on-policy RL methods (which leverage both positive and negative signals), potentially reducing the method's applicability to broader settings where learning from failures is critical.

W2: **Over-engineered pipeline for token efficiency.** If the core goal is token efficiency, the current pipeline may be unnecessarily complex: it requires a proprietary teacher model (GPT-4o), predefined template-based cognitive levels (with fixed structures like Goal, Evaluation, etc.), two-stage training (CoSFT + CoPO). A simpler alternative — such as adding a token-length penalty on thinking to the reward function — could achieve similar efficiency gains without the heavy scaffolding. This baseline is not compared against, making it hard to assess whether the complexity of CogRouter is justified.

W3: **Environment diversity is insufficient for a general-purpose method.** ALFWorld and ScienceWorld are both virtual, idealized environments with similar task structures. For a method that claims generality, evaluation on more realistic benchmarks is needed — e.g., WebShop (e-commerce interaction) or GDPVal (Industrial document generation). The current evaluation cannot distinguish whether CoPO's gains are specific to idealized settings or transferable to real-world scenarios.

---

> ### Author Rebuttal · Authors · 2026-03-30
>
> > **W1**: Lack of negative training signal.
>
> We would like to clarify that CoPO does utilize negative signals from failed trajectories. As shown in Equation 5 (Lines 258–271), failed trajectories ($i \in \mathcal{I}^{-}$) receive negative trajectory-level advantages and contribute to the standard GRPO-style policy update. The only difference is that their advantages are not reweighted by confidence scores (Equation 4, Lines 244–246).
> This design is intentional. Confidence-aware reweighting on failed trajectories would penalize cognitive levels based on how confidently they predict an incorrect action, which is a misleading signal. Our ablation (Table 2, `"w/ Corr & Incorr"`, Lines 430–439) confirms this: applying reweighting to both correct and incorrect trajectories degrades performance from 72.0% to 66.0% SR.
>
> > **W2**: Missing baseline with direct length penalty.
>
> Thank you for this insightful comment. We adopt the length reward from Kimi k1.5 [1] and ThinkDial [2]. The reward is defined as:
>
> $$R_i = R_i^{task} + \alpha \cdot R_i^{length}, \quad \text{where} \quad R_i^{length} = 0.5 - \frac{\mathrm{len}(i) - \mathrm{len\_{min}}}{\mathrm{len\_{max}} - \mathrm{len\_{min}}}$$
>
> Here $\mathrm{len\_{min}}$ and $\mathrm{len\_{max}}$ are computed within each trajectory group, requiring no predefined token budget. We vary $\alpha \in \\{0, 0.5, 1.0\\}$ to control the penalty strength. For the SFT stage, we use ReAct-style trajectories without cognitive level structure, with the same number of training samples as CoSFT. We then train GRPO with the above reward on ALFWorld using Qwen2.5-7B for 60 iterations.
>
> | Method | SR | #Tokens |
> |--------|-----|---------|
> | GRPO w/ length penalty ($\alpha=0$) | 67.5 | 3336.6 |
> | GRPO w/ length penalty ($\alpha=0.5$) | 63.0 | 2651.4 |
> | GRPO w/ length penalty ($\alpha=1.0$) | 56.5 | 2083.5 |
> | **CoSFT+CoPO (60 iter)** | **82.5** | **2031.6** |
>
> **Key findings:** Length penalty enforces trajectory-level compression, creating an accuracy-efficiency trade-off (SR drops 67.5% → 56.5% as penalty increases). In contrast, CoPO achieves both the highest SR (82.5%) and the lowest token usage (2031.6). This confirms that agentic tasks require **complexity-aware cognitive routing**: the ability to allocate deep reasoning to strategic steps while using fast responses for routine ones. Uniform length penalty cannot achieve this because it compresses all steps indiscriminately.
>
> We will run the full 150 iteration experiments including ScienceWorld and Llama3.1-8B, and incorporate these results as an additional baseline in the main results of the revised manuscript.
>
> [1] Kimi k1.5: Scaling Reinforcement Learning with LLMs
>
> [2] ThinkDial: An Open Recipe for Controlling Reasoning Effort in Large Language Models
>
> > **W3**: Environment diversity is insufficient.
>
> We appreciate this suggestion for broader evaluation. We conducted additional experiments on WebShop, an interactive web environment designed for web shopping, using Qwen2.5-7B as the base model. We use 6910 simulations from AgentGym [1]. For CoSFT, we randomly sample 500 environments to create a balanced dataset across four cognitive levels. For the RL stage, we train on the remaining environments for 60 iterations. Results are evaluated on 200 test simulations.
>
> | Method | Score | SR | #Tokens |
> |--------|-------|-----|---------|
> | Qwen2.5-7B (ReAct) | 46.2 | 19.5 | 1584.3 |
> | GRPO (60 iter) | 70.2 | 55.5 | 2917.4 |
> | **CoPO (60 iter)** | **75.4** | **62.0** | **1723.6** |
>
> **Key findings:** CoPO consistently outperforms GRPO on WebShop, achieving higher score (+5.2) and SR (+6.5%) while using 40% fewer tokens. This demonstrates that complexity-aware cognitive routing transfers effectively to realistic e-commerce scenarios.
>
> We will include full 150 iteration results in the main results of the revised manuscript.
>
> [1] AGENTGYM: Evolving Large Language Model-based
> Agents across Diverse Environments

---

> > ### Author Rebuttal · Reviewer_PvCE · 2026-04-04
> >
> > The authors have thoroughly addressed all three weaknesses with clarifications, new baselines, and additional experiments on WebShop. I maintain my very positive score of 5 (Accept).

---

### Official Review · Reviewer_Syb5 · 2026-03-16

**Soundness:** 2
**Presentation:** 1
**Significance:** 2
**Originality:** 3
**Overall Recommendation:** 3
**Confidence:** 2

**Summary:**

The paper aims to develop agents that adapt reasoning into four levels of depth based on confidence in solving. It utilizes a post-hoc style reasoning to train model reasoning, generating traces as justifications for already-fixed successful actions to instill stable cognitive patterns. COSFT builds supervised data by taking expert GPT-4o trajectories and attaching level-specific thought traces, and COPO, RL optimization for models to assign different reasoning depths to different states. The method is evaluated on ALFWorld and ScienceWorld, two text-based agent benchmarks requiring multi-step planning and interaction.

**Compliance With Llm Reviewing Policy:**

Affirmed.

**Final Justification:**

I kept my score because the main technical concern remains unresolved: both the supervision and RL signal rely on post-hoc thoughts generated conditioned on an already-fixed action. This makes it unclear whether the method truly learns the appropriate cognitive depth for a state, or mainly learns which rationale style best supports the target action under synthetic supervision. The rebuttal usefully argues against a pure verbosity effect. Given this issue, I still view the contribution as promising but not yet fully validated and will keep my score.

**Key Questions For Authors:**

Please correct me in rebuttal if this is wrong: You first collect "expert trajectories" from GPT-4o that consist only of observation-action pairs, then randomly assign a cognitive level and prompt the model to "complete the thinking process" that leads to that already-fixed ground-truth action, right ?
I understand the argument that this approach is necessary to instill stable, level-specific patterns and prevent "format leakage". But this is weak.

If I am reading the results correctly, COSFT with random balanced levels underperforms COSFTexp further suggests the annotation procedure itself strongly shapes the result. Models initialized with COSFTexp (where the teacher selects the level) actually performed worse after RL compared to the balanced random sampling ?  So the "expert" routing in the supervision phase determines the agent's adaptive capabilities


Also a general question/curiosity I had reading your work: how does hierarchy of decomposition effect the confidence/difficulty in solving the tasks? There would be an optimal hierarchy were all tasks are low difficulty and vice-versa? So a look-ahead planner if given similar supervision can solve this better?

**Limitations:**

not sure how to rate this, but not a great discussion of imitations.

**Strengths And Weaknesses:**

Strength

The paper aims to develop agents that adapt reasoning into four levels of depth based on confidence in solving. This is at some level is running into a version of the verification problem. They use confidence as supervision to tackle this which is interesting The link with cog-sci is also interesting.

Weakness

The paper devices method to let an agent adapt its reasoning depth step by step rather than think uniformly deeply or uniformly shallowly. The built on the core insight that the contextual utility of a reasoning depth can be measured by how confidently the model predicts the correct action after that reasoning. This assumes strong calibration of the model. The normalization using the 4 steps could mitigate it, but it also has a key issue.
COSFT collects successful observation–action trajectories from GPT-4o, then randomly assigns a cognitive level, and finally ask GPT-4o to generate a reasoning trace that leads to the already-fixed action. Copo again regenerates alternative thoughts for all four levels while keeping the observation and action fixed. So the “thinking” is not an observed latent process that produced the action; it is a post hoc explanation conditioned on the answer.
That weakens the claim that the model learned adaptive cognition, as opposed to adaptive verbosity or adaptive scaffold selection. This means the intermediate thought is partly a function of the target label, not just the state. In effect, the reasoning trace contains privileged supervision unavailable at inference time.
This also kind of makes the confidence-based RL signal somewhat circular. COPO scores each cognitive level by how confidently the model predicts the fixed action given the generated thought. But if the thought itself was generated to rationalize that action, then higher confidence may simply reflect how directly that thought encodes the action, not whether that level was truly appropriate for the state.

The evidence does not fully separate better routing from better synthetic supervision, though it partially covers up with comparison with GRPO. The reasoning is conditioned on the ground-truth action itself so the quality of gain is not clear. It applies RL that reweights step-level updates using how confidently the model predicts the fixed action under alternative level-conditioned thoughts. The reported gains on ALFWorld and ScienceWorld may therefore reflect the strength of this synthetic annotation and control scheme, rather than evidence that the model has learned natural or psychologically meaningful cognitive modes (though it is better than GRPO ). To separate “better routing” from “better synthetic supervision.” is challenging in this formulation beyond comparison with some methods that mode collapse. Is there a better ablation that can show this? The fact that COSFT with random balanced levels performs poorly before RL, while COSFTexp is better, already suggests the annotation scheme strongly shapes outcomes.

---

> ### Author Rebuttal · Authors · 2026-03-30
>
> > **W1**: Post-hoc thought generation makes the confidence signal circular.
>
> We appreciate this thoughtful concern. We address it as follows:
>
> **1. Balanced initialization eliminates format bias.** As mentioned in Lines 208-210, CoSFT ensures uniform distribution across all four cognitive levels, preventing the model from developing inherent preferences for any particular template format.
>
> **2. Confidence reflects context, not template properties.** If confidence were driven by post-hoc encoding capacity, longer templates should systematically produce higher confidence (L4 > L3 > L2 > L1). However:
>
> - *L1 dominates despite having no encoding capacity.* As shown in Figure 3, L1 accounts for 55.0% (ALFWorld) and 72.6% (ScienceWorld) of selections, using only the fixed string "Okay, I think I have finished thinking" with zero variable content. The action cannot be encoded in the thought at all. This indicates the state itself is **sufficient** for confident prediction on most steps.
> - *Allocation varies with context across L2/L3/L4.* Section 4.3 and Figure 5 show that L2 dominates early stages (26.3%) when parsing observations is critical, L4 peaks at initialization (21.9%) for global planning then drops to 6.8%, and L1 takes over in later stages for routine execution. This functional specialization is driven by state properties, not template length.
>
> **3. Relative confidence remains a valid signal.** Even if post-hoc completion introduces some noise, the *relative* differences across levels still indicate which cognitive depth best fits the current state. The context-sensitive patterns above confirm this: confidence captures whether a state requires deeper reasoning, not template verbosity.
>
>
> > **W2**: Gains may reflect stronger synthetic supervision rather than learned routing.
>
> We want to argue that the **performance gains arise from effective cognitive routing rather than data augmentation from cognitive group expansion**. CoSFT$\_{exp}$ undergoes identical cognitive group expansion during CoPO, generating the same four-level thinking variants at each step. However, its biased initialization (90.9% L3 after CoSFT$\_{exp}$, Figure 3) causes confidence reweighting to systematically favor L3 variants, and this bias persists through CoPO training (69.1% L3, Table 2), degrading routing quality. Despite identical data augmentation, CoSFT$\_{exp}$ performs worse after CoPO (62.0% vs. 72.0%). This demonstrates that routing quality, not data augmentation, drives CoPO's gains.
>
> > **Q1 & Q2**: The post-hoc annotation approach is weak; CoSFT$\_{exp}$ results suggest annotation determines adaptive capabilities.
>
> Yes, the understanding of CoSFT is correct. The balanced initialization is necessary. As we discuss in Lines 190–194, beyond preventing format collapse, it **eliminates the inherent cognitive level preferences of the base or expert model**. If such biases persist, the confidence signal during CoPO reflects learned format preferences rather than whether the current state actually requires that depth of reasoning.
> This is precisely what happens with CoSFT$\_{exp}$ initialization: the biased distribution (90.9% L3) persists through RL (69.1% L3 after training), causing confidence to systematically favor L3 regardless of context and preventing effective routing. Despite stronger SFT performance (78.5% CoSFT$\_{exp}$ vs 57.0% CogSFT on ALFWorld), CoSFT$\_{exp}$ performs worse after identical CoPO training (62.0% vs 72.0% on ScienceWorld). This shows that final adaptive capabilities are not determined by annotation quality, but by **whether the initialization allows unbiased routing learning during RL**.
>
> > **Q3**: Could a look-ahead planner with optimal task decomposition solve this better?
>
> We believe that step-level adaptation is fundamentally necessary and cannot be replaced by better upfront planning (look-ahead or optimal decomposition).
>
> - The key reason is **partial observability**: agentic tasks require environment interaction, and critical information only emerges during execution (e.g., object locations in ALFWorld). No upfront decomposition can anticipate these responses, so cognitive demand remains inherently dynamic per step.
>
> - Figure 5 directly supports this: even in later stages of a task, CogRouter does not converge entirely to L1; higher thinking levels (L2-L4) persist throughout execution, confirming that step difficulty fluctuates due to dynamic environment feedback rather than monotonically decreasing after initial planning.
>
> > **Limitation**: not sure how to rate this, but not a great discussion of limitations.
>
> We will add a dedicated limitation discussion in the revised manuscript, covering: (1) our method targets **agentic tasks** with multi-turn interaction and sparse rewards; single-turn tasks with fixed difficulty may benefit from token-level approaches instead; (2) the limited scope of environments and model scales. Extending to more environments and larger models is our future work.

---

> > ### Author Rebuttal · Reviewer_Syb5 · 2026-04-04
> >
> > Thank a lot for the reply. I gives more clarity.
> > I am not fully confident of the claim "confidence captures whether a state requires deeper reasoning". I understand that the signal are not driven by verbosity alone,  (even L1 is getting used). But the label-conditioned rationalization still weakens this claim. Even a short or fixed template can still be part of a pipeline where the thought is generated after the action is fixed, so the confidence score may still reflect compatibility with a post hoc explanation rather than being the right level.
> >
> > I will keep my score.

---

> > > ### Author Response · Authors · 2026-04-04
> > >
> > > We sincerely thank the reviewer for the continued and thoughtful discussion.
> > >
> > > We would like to offer a different perspective on this point. We argue that in our framework, **"compatibility with a post-hoc explanation" inherently acts as a robust proxy for "the right cognitive level."** Rather than being opposing concepts, they are fundamentally aligned.
> > >
> > > While the thought is generated post-hoc conditioned on the ground-truth action and current context (history, state, action), achieving high compatibility requires the reasoning structure to *coherently bridge* the context and the action. If a cognitive level does not match the context's complexity, the model is forced into unnatural reasoning, resulting in lower confidence:
> > >
> > > - **Complex Contexts (Penalty for Under-thinking):** If the history contains failed explorations, L3 (reflection) allows the model to naturally leverage these experiences to justify the correct action. L2 lacks this capacity, forcing a disjointed explanation. L3 thus yields higher compatibility, correctly reflecting the need for reflection.
> > >
> > > - **Simple Contexts (Penalty for Over-thinking):** When the next action is obvious, forcing the model into L4 (forward simulation) requires it to hallucinate unnecessary planning steps that fit poorly with a straightforward context. Here, L1 yields the highest compatibility because no complex reasoning is warranted.
> > >
> > > Therefore, post-hoc compatibility effectively **encodes the cognitive demands of the context**, and confidence reflects **the alignment between reasoning structure and context complexity**.
> > >
> > > **To provide evidence**, we analyze **the top-3 most frequent states** associated with each cognitive level when it is selected with the highest confidence **during training**:
> > >
> > > > | Level | Top Observations | Interpretation |
> > > > |---|---|---|
> > > > | L1 | "The door is now open" (40.2%), "The door is not open" (18.3%), "You focus on the [object]" (17.0%) | Simple environmental feedback requiring only reflexive actions |
> > > > | L2 | "You move to the [other room]" (34.9%), "You move to the hallway" (19.6%), "[circuit] anode/cathode connection state" (17.2%) | Arrival at new environments requiring situational awareness |
> > > > | L3 | "No known action matches that input" (82.0%), "Connections must specify..." (12.8%), "The [container] isn't open, can't see inside" (2.8%) | Error signals requiring reflection and plan revision |
> > > > | L4 |  "(Task Start)"  (57.7%), "This room is called the [room]..." (19.7%), "Inside the [container] is:..." (10.2%) | Task initialization requiring strategic planning |
> > >
> > > **Key findings**: L1 is selected for routine transitions (e.g., after a door opens, simply walking through). L2 for arriving at new locations needing assessment (e.g., next action is typically `look around`). L3 is overwhelmingly associated with error states (82.0%), requiring reflection. L4 is selected mainly at task start (57.7%), requiring strategic planning. These associations emerge entirely from confidence-based routing, yet they align precisely with the intended cognitive function of each level. This provides direct evidence that post-hoc compatibility captures genuine contextual demands rather than superficial template preferences. We will include this analysis in the revised manuscript.
> > >
> > > This perspective is further supported by **test-time results**: the model generates in a fully forward manner without ground-truth actions, yet CoPO substantially outperforms GRPO (which predominantly uses L4), and adaptive routing (92.5%) outperforms any fixed-level model (L1: 76.5%, L4: 86.5%, Appendix J), confirming that level selection is meaningful. Figure 5 also confirms that level distributions vary systematically across trajectory stages and task complexity at test time, further confirming that routing is driven by contextual demands.

---

### Decision · Program_Chairs · 2026-04-30

**Decision:**

Accept (regular)

**Comment:**

The paper proposes CogRouter, a framework for step-level cognitive depth adaptation in LLM agents, combining cognition-aware supervised fine-tuning and policy optimization to route among four reasoning levels during long-horizon interaction. Reviewers broadly agreed that the paper is well motivated, technically interesting, and empirically strong, especially in its framing of adaptive reasoning depth. I find the remaining concerns not sufficient to outweigh the overall strengths. Overall, I recommend acceptance.